# Denoising Diffusion via Image-Based Rendering

**Titas Anciukevičius** [1,2]   **Fabian Manhardt** [2]   **Federico Tombari** [2,3]   **Paul Henderson** [4]
[1] University of Edinburgh   [2] Google   [3] Technical University of Munich   [4] University of Glasgow
https://anciukevicius.github.io/generative-image-based-rendering

## Abstract

Generating 3D scenes is a challenging open problem, which requires synthesizing plausible content that is fully consistent in 3D space. While recent methods such as neural radiance fields excel at view synthesis and 3D reconstruction, they cannot synthesize plausible details in unobserved regions since they lack a generative capability. Conversely, existing generative methods are typically not capable of reconstructing detailed, large-scale scenes in the wild, as they use limited-capacity 3D scene representations, require aligned camera poses, or rely on additional regularizers. In this work, we introduce the first diffusion model able to perform fast, detailed reconstruction and generation of real-world 3D scenes. To achieve this, we make three contributions. First, we introduce a new neural scene representation, IB-planes, that can efficiently and accurately represent large 3D scenes, dynamically allocating more capacity as needed to capture details visible in each image. Second, we propose a denoising-diffusion framework to learn a prior over this novel 3D scene representation, using only 2D images without the need for any additional supervision signal such as masks or depths. This supports 3D reconstruction and generation in a unified architecture. Third, we develop a principled approach to avoid trivial 3D solutions when integrating the image-based rendering with the diffusion model, by dropping out representations of some images. We evaluate the model on several challenging datasets of real and synthetic images, and demonstrate superior results on generation, novel view synthesis and 3D reconstruction.

## 1 Introduction

Generative models of the 3D world learnt from 2D images are powerful tools that enable synthesising 3D content without expensive manual creation of 3D assets. They are also crucial for 3D reconstruction from sparse images. In particular, classical 3D reconstruction techniques like multi-view stereo (Seitz et al., 2006a; Schönberger et al., 2016) and more recent approaches like NeRFs (Mildenhall et al., 2020) can reconstruct a 3D scene from a dense set of images (typically at least 20). However, they are not able to reconstruct regions that are not observed in any of the input images. Even methods like PixelNeRF (Yu et al., 2021) that are designed to generalise across scenes still fail to render plausible details in unobserved regions, typically producing blurry outputs. To mitigate this issue, it is necessary to estimate a *posterior distribution* on 3D scenes, conditioned on one or more images. The posterior distribution assigns high probability to scenes that align with the content in the images, and that are also realistic in unobserved areas. Subsequently, this allows us to *sample* diverse plausible scenes from the posterior, instead of predicting a blurred *average* over all possible scenes.

Despite the importance of the task, so far generative models of real-world 3D scenes have remained elusive due to three challenges. First, real-world scenes are often large, or even unbounded, making it difficult to define a scene representation that can express the details that may be visible, yet also enables learning a generative model. For representations that do scale well, it is typically challenging to learn a prior over them (Müller et al., 2022; Barron et al., 2021), since their representation of 3D structure lacks generality across different spatial locations and scenes. Although some representations such as 3D voxels (Peng et al., 2020) make it simple to learn a prior as they interpret features consistently across different locations and scenes, these methods only represent a bounded 3D volume and allocate modelling capacity uniformly across a finite grid, regardless of the scene content.

A second challenge is that large datasets of real-world 3D scenes are scarce, since they are time-consuming and expensive to obtain (Müller et al., 2022). Thus, some methods aim to build a generative model of 3D scenes using only 2D images for training. While achieving great results for

the task of 3D generation, all these methods exhibit several limitations. First, some works rely on large-scale datasets where all objects are placed in a canonical pose (Anciukevičius et al., 2023). This is possible when training on synthetic, object-centric datasets, but that does not allow generating realistic scenes. Indeed, for real-world scenes, it is very difficult to define a single canonical frame of reference and align all scenes to this. Other works instead do not require canonicalized objects, but still can only operate on object-centric data. Moreover, commonly these approaches even require object masks, as they leverage bounded scene representations such as tri-planes, that only work within a predefined 3D volume. This again significantly restricts their generation capabilities, as these methods can only synthesize isolated 3D objects instead of complete scenes.

A third challenge is that it is difficult to sample from the true posterior distribution over real scenes with unbounded volumes, as opposed to a less-expressive marginal distribution. Existing approaches for unbounded 3D scene sampling commonly follow an "infer, fuse, render, and repeat" paradigm (Wiles et al., 2020). These sample parts of the scene visible in the 'next' camera view frustum conditioned on a small marginal observation of the current 3D scene (features or pixels of the scene projected into that image). However, they do not use information from *all* previously seen or generated images to predict a camera view frustum consistent with the complete scene.

In this work we propose, the **first denoising diffusion model that can generate and reconstruct large-scale and detailed 3D scenes**. To achieve this, we make the following technical contributions that respectively address each of the challenges above:

1. We introduce a new neural representation for unbounded 3D scenes, *IB-planes*, which increases expressiveness versus prior image-base rendering representations, by letting the model incorporate information from multiple images, and by adding additional depth and polar features.
2. We introduce a joint multi-view denoising framework incorporating a latent 3D scene. It supports unconditional generation and reconstruction of 3D from varying numbers of images; in both cases it samples from a true joint distribution over full 3D scenes, rather than a less-expressive marginal distribution.
3. We present the first principled approach for integrating image-based rendering into diffusion models: we drop out parts of the image-based scene representation corresponding to the view being rendered to prevent trivial 3D solutions, but introduce a cross-view-attentive architecture that enables the noise from all images to influence the latent 3D scene.

We evaluate our method on four challenging datasets of multi-view images, including CO3D (Reizenstein et al., 2021) and MVImgNet (Yu et al., 2023b). We show that our model *GIBR* (*Generative Image-Based Rendering*) learns a strong prior over complex 3D scenes, and enables generating plausible 3D reconstructions given one or many images. It outputs explicit representations of 3D scenes, that can be rendered at resolutions up to $1024^2$.

## 2 RELATED WORK

Traditional 3D reconstruction methods output scenes represented as meshes, voxels, or point-clouds (Schönberger & Frahm, 2016; Seitz et al., 2006b; Häne et al., 2013). Recently however, *neural fields* (Xie et al., 2022; Mildenhall et al., 2020) have become the dominant representation. These approaches represent a scene as a function mapping position to density and color; the scene is queried and rendered using volumetric ray marching (Max, 1995). That function may be a generic neural network (Park et al., 2019; Mildenhall et al., 2020; Barron et al., 2021), or a specifically-designed function (Peng et al., 2020; Fridovich-Keil et al., 2023b; Müller et al., 2022; Li et al., 2023; Chen et al., 2022; Xu et al., 2022) to improve performance. Due to their continuous nature, such representations are easily learnt from a dense set of images ($> 20$), by gradient descent on a pixel reconstruction loss. Some works allow reconstruction from fewer views (Yu et al., 2021; Wang et al., 2021; Chen et al., 2021; Liu et al., 2022; Henzler et al., 2021; Wiles et al., 2020; Liu et al., 2022; Wu et al., 2023), often by unprojecting features or pixels from the images into 3D space. However, typically parts of the scene will be unobserved (e.g. far outside the camera view frustum), and thus ambiguous or uncertain given the observed images. The methods above make a single deterministic prediction, and cannot synthesise details in unobserved regions; instead, they produce a blurred prediction corresponding to the mean over all possible scenes, without an ability to sample individual, plausible scenes. Other approaches incorporate ad-hoc losses or regularizers from pretrained generative models to improve realism of unobserved regions (Zhou & Tulsiani, 2023; Yoo et al., 2023; Zou et al., 2023;

Melas-Kyriazi et al., 2023; Liu et al., 2023a; Wynn & Turmukhambetov, 2023; Niemeyer et al., 2022), however no work has achieved a principled approach to generate samples of large-scale 3D scenes given one or more real images as input. In particular, methods based on score-distillation regularise scenes towards high-probability regions, but do not truly sample the distribution.

Generative models allow sampling from complex, high-dimensional distributions (e.g. a distribution of 3D scenes). A myriad of generative models have been proposed for different domains, including GANs (Goodfellow et al., 2014; Arjovsky et al., 2017; Karras et al., 2019), VAEs (Kingma & Welling, 2014; Van Den Oord et al., 2017), and autoregressive models (Van Den Oord et al., 2016). Diffusion models (Sohl-Dickstein et al., 2015) have recently outperformed their counterparts in most domains, including images (Kingma et al., 2021; Dhariwal & Nichol, 2021; Ho et al., 2022; Saharia et al., 2022; Lugmayr et al., 2022; Jabri et al., 2022), video (Blattmann et al., 2023), and music (Huang et al., 2023). Numerous works have trained diffusion models directly on classical (Luo & Hu, 2021; Vahdat et al., 2022; Chen et al., 2023; Zhou et al., 2021; Hui et al., 2022; Li et al., 2022; Cheng et al., 2023) and neural (Müller et al., 2022; Bautista et al., 2022; Wang et al., 2022b; Kim et al., 2023; Shue et al., 2022; Gupta et al., 2023; Karnewar et al., 2023; Gu et al., 2023) 3D scene representations. However, diverse, high-quality generation has remained elusive since such models are restricted by the lack of suitable datasets of canonically-oriented and bounded 3D scenes. In contrast, we aim to learn a generative 3D model from in-the-wild dataset of images (i.e. that could be easily collected with a camera and COLMAP pose estimation), without assuming canonical orientations, bounding boxes, object segmentations.

To mitigate the lack of 3D data, others methods use pretrained generative models of 2D images to guide the optimization of a 3D scene (Jain et al., 2022; Poole et al., 2022; Höllein et al., 2023; Fridman et al., 2023; Wang et al., 2023; 2022a; Metzer et al., 2022; Lin et al., 2023; Shi et al., 2023). However, such approaches do not scale to large scenes, nor allow posterior sampling of 3D scenes given one or more images as conditioning. An alternative approach is to learn a density jointly over 2D images and their latent 3D representations; this allows them to be trained from widely-available 2D image datasets, yet still sample 3D scenes (Skorokhodov et al., 2023; Xiang et al., 2023; Shi et al., 2022). Initially based on GANs (Chan et al., 2022; Deng et al., 2022; Nguyen-Phuoc et al., 2020; 2019; Schwarz et al., 2020; Zhao et al., 2022; Devries et al., 2021) or VAEs (Anciukevicius et al., 2022; Kosiorek et al., 2021; Henderson & Lampert, 2020; Henderson et al., 2020), recently diffusion-based methods have achieved the most promising results. Notably, (Anciukevičius et al., 2023) showed that diffusion can also perform 3D reconstruction by inferring a latent 3D representation given an image (unlike GANs), yet also generates sharp, detailed 3D assets and images (unlike VAEs). However, (Anciukevičius et al., 2023; Szymanowicz et al., 2023) are limited to object-centric and canonically-aligned scenes, due to their use of canonically-placed voxel grids or triplanes as 3D representations. Other works therefore uses a pipeline of "infer, fuse, render, and repeat" (Wiles et al., 2020) : the model generates the content visible in a camera view frustum conditioned on a rendering of the current scene at that frustum, then renders it to another viewpoint, and repeats. However, this only conditions on a marginal observation (since only the most recent view is seen, not the entire history of generated views nor an explicit 3D representation). Instead we aim to sample from a joint distribution of scenes. Moreover, they are slow to perform 3D reconstruction, e.g. concurrent work (Tewari et al., 2023) takes 2 hours, and do not support unconditional generation of 3D scenes nor conditional generation with arbitrary numbers of conditioning images.

Some works circumvent the difficulty of learning a 3D representation entirely by training conditional generative models to output images from novel viewpoints conditioned on one or more input images and a camera pose (Eslami et al., 2018; Kulhánek et al., 2022; Rombach et al., 2021; Du et al., 2023; Ren & Wang, 2022; Watson et al., 2022; Chan et al., 2023; Tseng et al., 2023; Liu et al., 2023b; Cai et al., 2022; Tang et al., 2023; Yu et al., 2023a). However, as such methods do not explicitly represent the underlying 3D scene, they cannot guarantee the resulting images depict a single consistent scene, and existing methods fail to generalize to camera poses far from the training distribution.

## 3 METHOD

Our goal is to build a generative 3D scene model that supports two tasks: (i) unconditional generation, (sampling 3D scenes a priori) (ii) 3D reconstruction (generation conditioned on one or more images). We aim to learn this model without 3D supervision by only assuming access to a dataset of multi-view images with relative camera poses (which can be easily obtained via structure-from-motion).

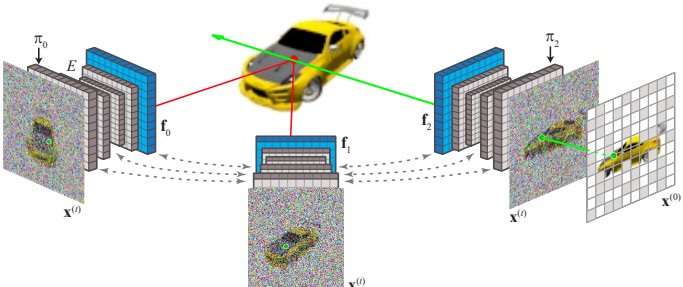

Figure 1: Our neural scene representation *IB-planes* defines 3D content using image-space features. Each camera $\pi_v$ is associated with a feature-map $\mathbf{f}_v$ (blue); together both parametrise a neural field that defines density and color for each 3D point $p$ (red dot). We incorporate this representation in a diffusion model over multi-view images. At each denoising step, noisy images $\mathbf{x}^{(t)}$ are encoded by a U-Net $E$ with cross-view attention (gray dashed arrows), that yields pixel-aligned features $\mathbf{f}_v$ (blue). To render pixels of denoised images (only one $\mathbf{x}^{(0)}$ is shown for clarity), we use volumetric ray-marching (green arrow), decoding features unprojected (red lines) from the other viewpoints.

To this end, we first describe a novel image-based 3D scene representation that adapts its capacity to capture all the detail in a set of images, yet is suitable for learning a prior over (Sec. 3.1). This enables us to define a denoising diffusion model over multi-view images depicting real-world scenes, that builds and renders an explicit 3D representation of the latent scene at each denoising step (Sec. 3.2). This ensures the generated multi-view images depict a single, consistent 3D scene, and allows rendering the final scene efficiently from any viewpoint. We name our model *Generative Image-Based Rendering* (GIBR).

## 3.1 REPRESENTING 3D SCENES WITH IB-PLANES

We represent a 3D scene as a neural field (Mildenhall et al., 2020) – a function mapping world-space positions to a density (i.e. opacity) and color, which can be rendered using the standard emission-absorption method (Max, 1995). Inspired by recent success of image-based rendering (Lensch et al., 2003; Yu et al., 2021) and K-planes (Fridovich-Keil et al., 2023a), the density and color at each 3D point are defined via features placed in the view space of a set of images (Fig. 1). Specifically, we represent a scene by a set of 2D feature-maps $\{\mathbf{f}_v\}_{v=1}^V$ and corresponding poses $\{\pi_v\}_{v=1}^V$ for $V$ cameras. These per-view feature-maps and poses parametrize a single neural field that defines the density and color at each point $p \in \mathbb{R}^3$ in 3D space. To calculate these, we project $p$ into each camera view based on its pose $\pi_v$ (which includes both extrinsics and intrinsics), finding the corresponding pixel-space location $\phi(p, \pi_v)$. Then, we extract the feature vector at that location in $\mathbf{f}_v$ using bilinear interpolation, setting $f_v(p) = \mathbf{f}_v[\phi(p, \pi_v)]$.

Notably, unlike PixelNeRF (Yu et al., 2021), our IBR feature planes (which we name *IB-planes*) are output jointly by a U-Net that attends over multiple views. Hence, IB-planes are strictly more expressive than prior IBR approaches, such as PixelNeRF and IBRNet (Wang et al., 2021), that calculate features independently for each image. This is because the multi-view U-Net can arrange different IBR features for a viewpoint depending on other input images, and remove the depth ambiguity that is present when given only one image. On the other hand, unlike K-planes (Fridovich-Keil et al., 2023a), our IB-planes are placed in the camera view frusta to facilitate learning a generalizable model that maps images to scene representations. As a result, we can use a simple and fast max-pooling operation to fuse features, instead of needing a large, expensive feature-fusion model (e.g. IBRNet has a deep attention network over point features and nearby 3D points).

To ensure the scene geometry is well-defined outside the union of the camera view frusta, for each camera we also calculate a polar representation of the displacement of $p$ relative to the camera's center, and use the resulting angles to interpolate into a second feature map (with an equirectangular projection), giving a vector $f_v'(p)$. We concatenate the feature vectors $f_v(p)$ and $f_v'(p)$ with an embedding of the distance of $p$ from the corresponding camera origin, and process this with an MLP to give a feature vector $f_v^*(p)$. We next max-pool these feature vectors across views, to give a single unified feature $f(p) = \max_v f_v^*(p)$ that fuses information from all views; the $\max$ is computed element-wise. Finally, this is mapped by an MLP to the density and RGB color at $p$.

## 3.2 MULTI-VIEW DENOISING DIFFUSION

We next describe our generative model of multi-view images, then discuss how we incorporate our scene representation into this to ensure 3D consistency while retaining expressiveness. We want to learn a generative model over sparse multi-view images $\mathbf{x}^s$ drawn from some unknown distribution $\mathcal{X}$, where each $\mathbf{x}^s \in \mathbb{R}^{V \times H \times W \times 3}$ depicts a different scene, and consists of $V$ RGB images each of size $W \times H$ (note that $V$ may vary between scenes). Associated with each view $x_v^s$ is a camera pose $\pi_v^s$, specified relative to $x_0^s$ (i.e. we do *not* assume existence of a canonical coordinate system common to all scenes, unlike e.g. Anciukevičius et al. (2023) and Chan et al. (2022). In the following description we omit the scene index $s$ for clarity.

In order to define a generative model over multi-view images $\mathbf{x}$, we define forward (noising) and reverse (denoising) diffusion processes (Ho et al., 2020). The forward process is a sequence of stochastic transformations that progressively add Gaussian noise to the original pixels, resulting in a unit Gaussian sample over time. Formally, for a time step $t$ and noise level $\beta_t$ determined by a predefined noise schedule, the noisy multi-view image at diffusion time step $t$ is:

$$\mathbf{x}^{(t)} = \sqrt{1 - \beta_t}\,\mathbf{x}^{(t-1)} + \sqrt{\beta_t}\,\epsilon^{(t)}, \quad \epsilon^{(t)} \sim \mathcal{N}(\mathbf{0}, I) \tag{1}$$

To sample from the original distribution $\mathcal{X}$, we learn a reverse process that reconstructs multi-view images from their noised versions. Specifically, we train a denoising function $\boldsymbol{\mu}_\theta(\mathbf{x}^{(t)}, t)$ to predict the original multi-view image $\mathbf{x}$ from the noisy image $\mathbf{x}^{(t)}$ and the diffusion step $t$ (note we predict the image, not the noise as is common). To sample new multi-view images, we begin from a sample of pure Gaussian noise, and repeatedly apply $\boldsymbol{\mu}_\theta$ following the DDIM sampler of Song et al. (2020).

Typically diffusion models implement $\boldsymbol{\mu}_\theta$ as a neural network, often a U-Net (Ronneberger et al., 2015). This could be applied in our multi-view setting, provided we allow different views to exchange information, e.g. using a 3D U-Net, or cross-attention between the views. However, it does not *guarantee* that the resulting images are 3D-consistent, i.e. that the same 3D scene is visible in each view – the model must instead learn to approximate this, and often fails (see our ablation study). We next describe a denoiser $\boldsymbol{\mu}_\theta$ that ensures the views are 3D-consistent throughout the diffusion process.

**3D-consistent denoising.** To ensure 3D consistency of the multi-view images reconstructed during the diffusion process, and to enable access to a 3D model of the final scene, we incorporate an explicit intermediate 3D representation into the architecture of our multi-view denoiser $\boldsymbol{\mu}_\theta$. During each denoising step, an encoder $E$ estimates a single noise-free 3D scene $\{(\mathbf{f}_v, \pi_v)\}_{v=1}^V = E(\mathbf{x}^{(t)}, t)$ parametrized according to Sec. 3.1 that incorporates information from all the views. The denoiser then renders this scene from each viewpoint to yield the denoised views, so we have

$$\boldsymbol{\mu}_\theta(\mathbf{x}^{(t)}, t) = \text{render}\left(E(\mathbf{x}^{(t)}, t)\right). \tag{2}$$

**Setwise multi-view encoder.** The encoder $E(\mathbf{x}^{(t)}, t)$ calculates pixel-aligned features $\mathbf{f}_v$ for each view $\mathbf{x}_v^{(t)}$ in $\mathbf{x}^{(t)}$ using a multi-view U-Net architecture. We adapt the U-Net architecture of (Ho et al., 2020), modifying the output layer yield features instead of RGB values. We also introduce attention between views, allowing them to exchange information. We replace each attention layer with a multi-headed linear attention (Vaswani et al., 2017; Katharopoulos et al., 2020) that jointly attends across all feature locations in all views. Aside from these attention layers, the rest of the network processes each view independently; this is more computationally efficient than a full 3D CNN. It also avoids any undesirable inductive bias toward smoothness across adjacent views, which is important since we do not assume views have any particular spatial relation to each other. We also provide the encoder with a setwise embedding of the camera poses $\pi_v$, specified relative to some arbitrary view. We flatten the extrinsics and intrinsics matrices to vectors, pass them to small MLPs, and concatenate the results, to give a per-view relative pose embedding $\pi_v^*$. When encoding each view $\mathbf{x}_v^{(t)}$, we input the corresponding embedding $\pi_v^*$, and also the result of max-pooling the embeddings for other views. This is injected into the network similarly to the Fourier embedding of the timestep $t$, by concatenating it with the features at each layer. Importantly, our encoder architecture jointly reasons over all images in the scene; unlike autoregressive methods (e.g. Wiles et al., 2020), all information from all views is accounted for simultaneously to ensure the scene is coherent. Moreover, use of pooling operations in the encoder and the scene representation (feature fusion) to integrate information from different views ensures that it supports varying numbers of images.

**Conditional generation.** We can adapt this model to the conditional setting, where we are provided with one or more input views and must generate complete scenes. In this case, some views passed to

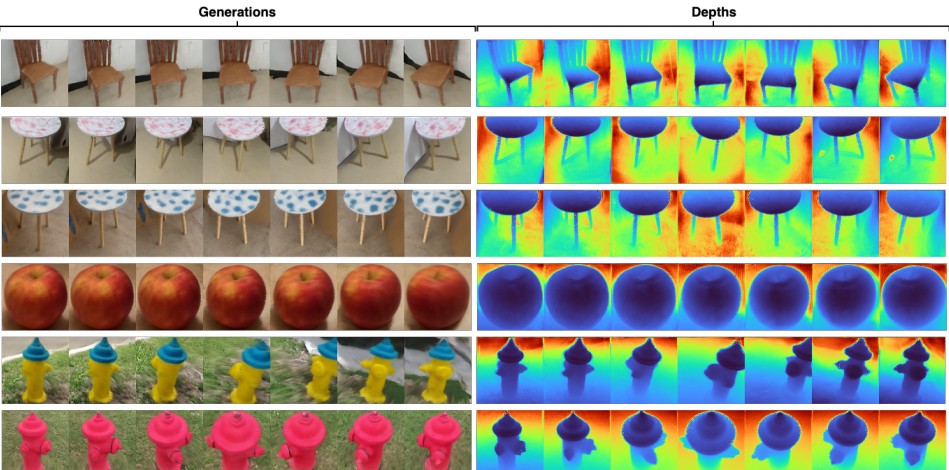

Figure 2: Samples generated by our method trained on MVImgNet (first three rows), CO3D (last three rows). Note that each multi-view image depicts a single coherent scene, with plausible appearance and detailed geometry. Please see the supplementary material for $1024 \times 1024$ video visualisations.

$\boldsymbol{\mu}_\theta$ as part of $\mathbf{x}^{(t)}$ are not noisy. The $V$ views are therefore split into $V_n$ noisy views, and $V_c$ noise-free conditioning views. We indicate this to the model by passing a different $t$ for each view, with $t = 0$ indicating a noise-free view. Each noisy view then encodes (in its noise) latent information about parts of the scene that are uncertain even given the noise-free conditioning views. We ensure there is at least one noisy view present, so the model always retains generative behavior. The image-based scene representation ensures there is a direct flow of information from noise at the pixels to corresponding points in the 3D scene, while the joint multi-view encoder means that latent information is correctly fused across different views, also incorporating information from the observed images.

## 3.3 TRAINING

Our model is trained to reconstruct multi-view images $\mathbf{x}$ given their noised versions $\mathbf{x}^{(t)}$. We use an unweighted diffusion loss $\mathcal{L}$ (Ho et al., 2020) with an L1 photometric reconstruction term:

$$\mathcal{L} = \mathbb{E}_{t, \mathbf{x}} ||\mathbf{x} - \boldsymbol{\mu}_\theta(\mathbf{x}^{(t)}, t)||_1 \qquad (3)$$

We train our model end-to-end to minimize $\mathcal{L}$ using Adam (Kingma & Ba, 2015). We vary $V$ across different minibatches to ensure generality; to allow conditioning on varying numbers of images, we also vary the number $V_c$ of noise-free views between zero and $V$. Training the model to reconstruct a large number of high-resolution images is computationally expensive since it requires a volumetric ray-marching for $V \times H \times W$ pixels. To overcome this, we approximate the loss (3) by only rendering a small fraction ($\approx 5\%$) of rays. This is still an unbiased estimate of $\mathcal{L}$, and has surprisingly minimal effect on the number of iterations until convergence, while greatly improving the wall-clock time and allowing us to go beyond prior works by training at $256 \times 256$ resolution.

## 3.4 DROPPING OUT NEURAL REPRESENTATIONS

One major challenge with 3D-aware generative models is that minimizing the loss does not necessarily force the model to accurately understand 3D. The model can instead produce a simple, uninformative pseudo-3D representation, such as a flat plane positioned directly in front of each camera, textured with a projection of the observed scene from that angle. Recent techniques have tried to address this by using various dataset-specific approaches, like requiring camera poses in a canonical frame of reference (Anciukevičius et al., 2023) (which is not possible for in-the-wild scenes). A naïve approach would be to use held-out views for supervision, but this falls short as they prevent the diffusion model from sampling interpretations of these heldout views, instead merely approximating the average observation, much like older non-generative techniques. Instead, we adopt a principled approach that ensures an expressive 3D representation with purely the diffusion loss (3), without any regularizers, heldout views or canonical camera poses. Specifically, we drop out the features $\mathbf{f}_v$ from the $v^{\text{th}}$ view when rendering to that same viewpoint. Note that this is not the same as *masking* some noises (as previous methods did), since we still allow latent information in the noise of the $i^{\text{th}}$ view to flow to all other views' features and thus the scene itself. During inference, we include features from all views.

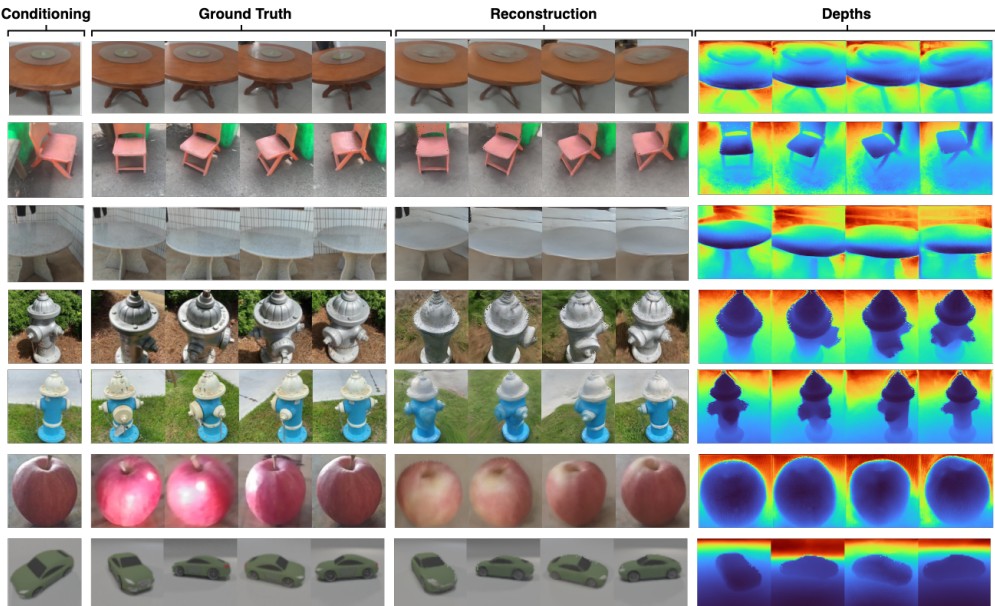

Figure 3: Results from our model on 3D reconstruction from a single image on MVImgNet (first 3 rows), CO3D (next 3 rows) and ShapeNet (last row). The leftmost column is the input; the next four show the ground-truth novel view images. The remaining columns show our model's prediction from those viewpoints and the predicted depth-maps. Please see the supplementary videos for more results.

## 4 EXPERIMENTS

**Datasets.** We evaluate our approach on three datasets: (i) real-world chairs, tables and sofas from MVImgNet (Yu et al., 2023b); (ii) real-world hydrants, apples, sandwiches and teddybears from CO3D (Reizenstein et al., 2021); (iii) the renderings of ShapeNet (Chang et al., 2015) cars from (Anciukevičius et al., 2023). For CO3D, we train single-class models for hydrant and apple, and also a class-conditional model over the four classes; for MVImgNet we train one class-conditional model. Notably, CO3D and MVImgNet show large-scale indoor and outdoor scenes, including objects with fine details and textures. For all datasets, we only use the RGB images and relative camera poses – we do *not* use any masks or depths. During training, we randomly sample 6–8 views per scene. For MVImgNet and CO3D, the images are resized to $96 \times 96$ for most experiments and $256 \times 256$ for high-resolution runs (only supported by our method); for ShapeNet we use the original $64 \times 64$. For CO3D, prior to resizing, we take a square crop centered on the ground-truth object mask; for MVImgNet, we take a center crop with size equal to the smaller dimension of the image.

**Baselines.** We compare to the most related diffusion method RenderDiffusion (Anciukevičius et al., 2023) and non-generative method PixelNeRF (Yu et al., 2021); the concurrent Viewset Diffusion (Szymanowicz et al., 2023); and the score-distillation method SparseFusion (Zhou & Tulsiani, 2023). Like ours, RenderDiffusion and Viewset Diffusion perform diffusion in image space. The former uses a triplane representation of 3D shapes and requires scenes to be placed in a canonical world-space, while the latter uses a fixed-size voxel grid. Thus, neither is able to adapt their capacity nor model very large scenes. Hence, we extend them to support our setting, and denote them as RenderDiffusion++, PixelNeRF++ and VSD*. Further details on how we extend them to our setting are in App. D.

### 4.1 GENERATIVE 3D RECONSTRUCTION

We first evaluate performance on 3D reconstruction from one or few images. We measure PSNR, SSIM and LPIPS between predicted and ground-truth images, and the rank-correlation of depths (DRC) (we use rank-correlation since absolute scale may differ between ground-truth and predicted scenes). Reconstruction from sparse images is ambiguous – there are many plausible completions of unobserved regions. We therefore follow other works on stochastic prediction (e.g. Denton & Fergus, 2018) and draw multiple (8) samples from the model, calculate the metrics for each, and take the best sample with respect to the ground-truth. For the diffusion-based methods, we render images and calculate metrics for two sets of viewpoints – the views in which the diffusion was performed (with

| | Single-view reconstruction | | | | | | | | Multi-view reconstruction | | | |
|---|---|---|---|---|---|---|---|---|---|---|---|---|
| | PSNR$_D$↑ | SSIM$_D$↑ | LPIPS$_D$↓ | DRC$_D$↑ | PSNR$_H$↑ | SSIM$_H$↑ | LPIPS$_H$↓ | DRC$_H$↑ | PSNR$_D$↑ | SSIM$_D$↑ | LPIPS$_D$↓ | DRC$_D$↑ |
| **CO3D hydrant** | | | | | | | | | | | | |
| RenderDiff++ | 15.70 | 0.317 | 0.598 | **0.832** | 16.28 | 0.333 | 0.587 | **0.837** | 18.60 | 0.399 | 0.533 | **0.882** |
| PixelNeRF++ | 15.06 | 0.278 | 0.615 | 0.527 | – | – | – | – | 16.86 | 0.366 | 0.545 | 0.595 |
| Viewset Diffusion | 13.18 | 0.144 | 0.714 | – | 13.50 | 0.149 | 0.718 | – | – | – | – | – |
| SparseFusion | 12.06 | – | 0.630 | – | – | – | – | – | – | – | – | – |
| **GIBR (ours)** | **16.07** | **0.329** | **0.456** | 0.821 | **17.12** | **0.403** | **0.449** | 0.829 | **20.22** | **0.571** | **0.283** | **0.882** |
| **CO3D apple** | | | | | | | | | | | | |
| RenderDiff++ | 16.71 | 0.601 | 0.475 | 0.708 | 17.20 | 0.608 | 0.464 | 0.730 | 18.97 | 0.638 | 0.427 | 0.648 |
| PixelNeRF++ | 16.25 | 0.546 | 0.548 | 0.513 | – | – | – | – | 17.73 | 0.601 | 0.476 | 0.542 |
| Viewset Diffusion | 13.99 | 0.416 | 0.633 | – | 13.31 | 0.393 | 0.674 | – | – | – | – | – |
| **GIBR (ours)** | **18.09** | **0.616** | **0.396** | **0.739** | **18.92** | **0.647** | **0.372** | **0.743** | **21.04** | **0.712** | **0.296** | **0.746** |
| **CO3D multi-class** | | | | | | | | | | | | |
| RenderDiff++ | 15.94 | 0.314 | 0.686 | 0.836 | 16.52 | 0.324 | 0.676 | 0.843 | 17.81 | 0.356 | 0.643 | 0.848 |
| PixelNeRF++ | 15.62 | 0.303 | 0.655 | 0.580 | – | – | – | – | 17.25 | 0.394 | 0.572 | 0.640 |
| **GIBR (ours)** | **16.70** | **0.360** | **0.481** | **0.863** | **17.90** | **0.434** | **0.465** | **0.872** | **21.54** | **0.634** | **0.281** | **0.898** |
| **ShapeNet car** | | | | | | | | | | | | |
| RenderDiff++ | 25.50 | 0.802 | 0.266 | 0.660 | 25.31 | 0.792 | 0.267 | 0.720 | 26.89 | 0.850 | 0.245 | 0.790 |
| PixelNeRF++ | 26.81 | 0.860 | 0.218 | 0.889 | – | – | – | – | 25.69 | 0.848 | 0.226 | 0.827 |
| Viewset Diffusion | 28.00 | 0.871 | 0.167 | – | 26.06 | 0.817 | 0.227 | – | – | – | – | – |
| **GIBR (ours)** | **29.74** | **0.906** | **0.139** | **0.993** | **28.96** | **0.883** | **0.162** | **0.992** | **33.46** | **0.961** | **0.096** | **0.998** |
| **MVImgNet furniture** | | | | | | | | | | | | |
| RenderDiff++ | 17.37 | 0.468 | 0.622 | – | 18.11 | 0.483 | 0.610 | – | 18.44 | 0.487 | 0.601 | – |
| PixelNeRF++ | 16.57 | 0.412 | 0.582 | – | – | – | – | – | 15.71 | 0.350 | 0.647 | – |
| Viewset Diffusion | 17.58 | 0.409 | 0.540 | – | 18.02 | 0.434 | 0.530 | – | – | – | – | – |
| **GIBR (ours)** | **18.54** | **0.518** | **0.414** | – | **19.89** | **0.590** | **0.369** | – | **22.09** | **0.730** | **0.284** | – |

Table 1: Results on 3D reconstruction from single and multiple images, for our method GIBR and baselines. Metrics suffixed D are calculated on the same views as we perform diffusion in; metrics suffixed with H are calculated in other, held-out views (except for PixelNeRF, which does not make this distinction). Note ground-truth depths are not available for MVImgNet, and Viewset Diffusion cannot perform reconstruction from six views. The SparseFusion result is from (Tewari et al., 2023).

| | GIBR (ours) | | RenderDiff++ | | VSD* |
|---|---|---|---|---|---|
| | FID$_D$↓ | FID$_H$↓ | FID$_D$↓ | FID$_H$↓ | FID$_D$↓ |
| CO3D hydrant | **91.9** | **118.1** | 185.4 | 182.9 | 217.0 |
| CO3D apple | **50.5** | **51.8** | 149.2 | 148.9 | 101.7 |
| CO3D multi | **121.5** | **123.4** | 201.6 | 201.2 | – |
| ShapeNet car | 62.8 | **90.1** | 163.5 | 160.2 | **56.0** |
| MVImgNet | **99.8** | **107.3** | 234.1 | 232.1 | 191.4 |

(a)

| | Generation | | Single-view reconstruction | | | |
|---|---|---|---|---|---|---|
| | FID$_D$↓ | FID$_H$↓ | PSNR$_D$↑ | SSIM$_D$↑ | LPIPS$_D$↓ | DRC$_D$↑ |
| CO3D hydrant | 183.8 | 194.7 | 15.10 | 0.320 | 0.636 | 0.785 |
| MVImgNet | 194.5 | 202.0 | 17.96 | 0.554 | 0.519 | – |

(b)

Table 2: **(a)** Results on generation for our method and two baselines. **(b)** Results on generation and 3D reconstruction for our method on high-resolution images ($256 \times 256$).

subscript D on the metric names), and a disjoint set of held-out viewpoints (subscript H). The latter show whether methods generate consistent 3D geometry that can be viewed from any angle.

Note that in App. A.2 we measure the impact of training our model with different numbers of views. Additional qualitative results are presented in App. A.3.

**Reconstruction from a single image.** We first evaluate reconstruction from one input image with unknown camera pose, meaning there is a high degree of uncertainty in the resulting scene, since much of it is unobserved. Quantitatively, our model GIBR out-performs both the recent generative 3D diffusion model RenderDiffusion (Anciukevičius et al., 2023), and the non-probabilistic PixelNeRF (Yu et al., 2021), across all datasets in terms of PSNR, SSIM, LPIPS and DRC ('single-view reconstruction' columns in Tab. 1). We attribute this to GIBR's generative capabilities (in contrast to deterministic PixelNeRF that must make blurry, averaged predictions), and to its flexible image-based scene representation (in contrast to RenderDiffusion which relies on fixed-size triplanes). Qualitative results (Fig. 3) confirm that not only does our model successfully reconstruct sharp and visually convincing 3D scenes, but it also excels at generating plausible details in regions that are not visible in the input view. The depth-maps show that even fine details such as chair legs are accurately captured. In Tab. 2b we evaluate our model on higher resolution images than supported by prior works ($256 \times 256$), showing that it retains competitive performance even in this more challenging setting, particularly on the multi-class MVImgNet dataset. Moreover, in the supplementary material, we show renderings of our reconstructed scenes at an even higher resolution ($1024 \times 1024$), which is only possible as our IB-planes representation explicitly captures the latent 3D scene.

**Reconstruction from multiple images.** Next, we evaluate performance on 3D reconstruction from six views. We see (Tab. 1, right four columns) that GIBR successfully makes use of the additional information in the larger number of conditioning images to improve the quantitative results versus reconstruction from a single image. This is akin to single-scene overfitting methods such as NeRF (though still with fewer images than they typically require), but still leverages our multi-view denoising U-Net architecture to ensure the scene remains close to the learnt prior distribution. Qualitative results are shown in Fig. 4 in the appendix; we see that while our model makes use of its learnt prior to complete unobserved regions, it still faithfully integrates the detailed texture and geometry visible in all observed viewpoints to reconstruct a coherent scene.

## 4.2 UNCONDITIONAL GENERATION OF 3D SCENES

We now evaluate performance on unconditional generation of 3D scenes. We measure performance with two variants of Fréchet Inception Distance (Heusel et al., 2017). $FID_D$ is calculated using renderings at the viewpoints at which diffusion was performed, i.e. the exact multi-view images output by the diffusion model. $FID_H$ instead uses renderings of the generated 3D shapes from seven different viewpoints, verifying that the 2D diffusion process yields a valid 3D shape (not just plausible projections in the views where the diffusion was performed).

Our model demonstrates significant improvements over both baselines according to $FID_D$ (Tab. 2a). Notably, our 3D generated scenes also look plausible from different viewpoints than those in which the model performed the denoising, as shown by the comparable values of $FID_H$ and $FID_D$. Concurrent Viewset Diffusion (Szymanowicz et al., 2023) performs worse on CO3D and MVImgNet, due to its use of a finite grid of features to represent the scene, meaning it must trade off detail for scene size; however it is the top-performing method on ShapeNet (which is simpler since objects and cameras are placed in a canonical frame of reference). Qualitatively (Fig. 2), our model not only generates visually coherent 3D scenes due to its explicit 3D representation, but also exhibits convincing 3D geometry, as seen in the crisp depth maps. We attribute this in part to our lack of restrictive regularisers, and in part to our expressive 3D representation and multi-view U-Net architecture, which together ensure the latent pixel noise in image space is integrated into a coherent 3D scene during the diffusion process. Further qualitative results (including from the baselines) and ablations are given in App. A.

## 4.3 ABLATION EXPERIMENTS

We performed five ablation experiments to quantify the benefit of our key technical contributions and design decisions, showing decreased performance of our model (i) without dropout of representation described in Sec. 3.3; (ii) replacing our IB-planes representation (Sec. 3.1) with triplanes; (iii) without cross-view attention; (iv) replacing volumetric rendering with a black-box 2D CNN; (v) without polar features. We report results on CO3D hydrant in Tab. 3 and discuss them in detail in App. A.4.

| | Generation | | Single-view reconstruction | | | | Multi-view reconstruction | | | |
|---|---|---|---|---|---|---|---|---|---|---|
| | $FID_D\downarrow$ | $FID_H\downarrow$ | $PSNR_D\uparrow$ | $SSIM_D\uparrow$ | $LPIPS_D\uparrow$ | $DRC_D\uparrow$ | $PSNR_D\uparrow$ | $SSIM_D\uparrow$ | $LPIPS_D\uparrow$ | $DRC_D\uparrow$ |
| No repr. drop. | 58.9 | 266.5 | 15.47 | 0.279 | 0.450 | 0.586 | 19.73 | 0.497 | 0.311 | 0.700 |
| No IBR | 176.4 | 177.9 | 14.55 | 0.273 | **0.631** | 0.782 | 17.39 | 0.349 | 0.569 | 0.839 |
| No cross-view attn. | 98.0 | 126.1 | 14.91 | 0.288 | 0.482 | 0.808 | 19.50 | 0.545 | 0.307 | 0.871 |
| No 3D | **36.3** | – | 13.51 | 0.186 | 0.509 | – | 14.04 | 0.208 | 0.472 | – |
| No polar features | 113.2 | 126.5 | **16.27** | **0.345** | 0.482 | 0.747 | **20.46** | **0.587** | **0.292** | 0.854 |
| Full model | 91.9 | **118.1** | 16.07 | 0.329 | 0.456 | **0.821** | 20.22 | 0.571 | 0.283 | **0.882** |

Table 3: Ablation results for variants of our method on CO3D hydrant. See App. A.4 for more details.

## 5 CONCLUSION

We have introduced a new approach to 3D scene generation and reconstruction, that can be trained from multi-view images without 3D supervision. Our denoising diffusion model *GIBR* incorporates an explicit 3D representation of the latent scene at each denoising step, ensuring that the resulting multi-view images always depict a single consistent 3D scene. To enable this, we introduced a powerful new scene representation based on image features lifted into 3D space, that can adapt its capacity according to the parts of the scene that are imaged, ensuring details are captured faithfully.

**Limitations.** While this work makes progress towards unsupervised learning of 3D generative models from in-the-wild images, it still assumes each scene is static. Also, even with approximation of loss (3), our model is slower to train than 2D diffusion models as it requires volumetric rendering.

## ACKNOWLEDGEMENTS

TA thanks Hakan Bilen, Christopher K. I. Williams, Oisin Mac Aodha, Zhengqi Li and Ben Poole for valuable feedback and fruitful discussions throughout the project. The authors also thank Michael Niemeyer and Michael Oechsle for proof-reading the paper. PH was supported in part by the Royal Society (RGS\R2\222045). TA was supported in part by an EPSRC Doctoral Training Partnership.

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

# A    ADDITIONAL EXPERIMENTS

## A.1    ABLATION EXPERIMENTS

We performed five ablation experiments to quantify the benefit of our key technical contributions and design decisions. We report results on the CO3D hydrant dataset in Tab. 3.

**No representation dropout.**  We first remove the neural representation dropout (No repr. drop.), i.e. when rendering from each view, we also include features from that view even during training. For generation, this results in a large performance degradation wrt $FID_H$ (266.5 vs 118.1), but an improvement wrt $FID_D$. This is because the ablated model learns trivial solutions, typically a plane placed directly in front of each camera, textured with a projection of the scene. Thus, each diffused viewpoint yields a visually-pleasing image, but these do not depict a single 3D-consistent scene; when we move to held-out viewpoints realism drops dramatically (i.e. higher $FID_H$). Depth accuracy for 3D reconstruction is also much poorer (e.g. 0.586 vs 0.821), since the predicted depths are not meaningful.

**No image-based representation.**  We next experiment with parameterizing the 3D scene using triplanes, instead of using image-aligned features. In this case the U-Net outputs three planes of features, which are mapped to a single volume of 3D space placed relative to the first camera (Chan et al., 2022; Chen et al., 2022); thus there is no longer a direct geometric correspondence between pixel features and the part of the scene they parameterize.  This leads to a substantial drop in model performance across all metrics, indicating that image-based representation is better able to model the details in the 3D scene. We attribute this to its ability to allocate capacity efficiently in the 3D volume, and to allow information flow directly from the noise at pixels to the underlying regions of the 3D scene.

**No cross-view attention.**  We remove our encoder's ability to attend across jointly across multiple views, by replacing the cross-view attention operation with a traditional attention operation that operates within each image independently. This results in a small drop in performance across all metrics, since only the local (per-3D-point) decoder MLP is available to integrate information across views.

**No polar features.**  We remove the polar unprojection of features, so the scene is defined only within the union of the view frustra of the cameras where diffusion was performed.  This results in a performance drop wrt $FID_H$, since it is now possible that parts of the scene visible in held-out viewpoints are undefined.  Interestingly, the performance on 3D reconstruction improves slightly; this may be because more modelling capacity is available for the central foreground region of the scene.

**No 3D.**  We next experiment with generating multi-view images using only 2D multi-view diffusion, still with cross-view attention so the model is expressive enough to learn 3D consistency, but without any explicit 3D scene representation. This model uses the same U-Net architecture, but the last layer of the U-Net directly outputs RGB images, instead of features for rendering. While the resulting images score highly in terms of realism according to the FID metric, they fail to maintain a coherent 3D scene. In other words, each image in the multi-view set appears to represent a different scene, even though each individual image looked convincing. This is quantitatively reflected in the low performance scores for novel view synthesis tasks, demonstrating the model's inability to accurately create new viewpoints of the scene, as confirmed by low PSNR metrics.

## A.2    VARYING THE NUMBER OF VIEWS

We now experiment with training our model using different numbers of views per scene (3–6) in each minibatch, instead of the default eight (Tab. 4). We use the same number of views for diffusion during testing as training, and therefore only report metrics on eight held-out views, to ensure the metrics are comparable across runs. We observe a significant improvement in unconditional generation performance ($FID_H$) as the number of views increases; performance is not saturating at eight views, so we hypothesise that further improvements would be possible simply by training with more views per scene (at the cost of increased computational expense). Reconstruction performance however appears to saturate around 5–6 views.

| #views | Generation | Single-image reconstruction | | | |
|---|---|---|---|---|---|
| | FID$_H\downarrow$ | PSNR$_H\uparrow$ | SSIM$_H\uparrow$ | LPIPS$_H\uparrow$ | DRC$_H\uparrow$ |
| 3 | 224.0 | 16.84 | 0.384 | 0.546 | 0.732 |
| 4 | 185.2 | 17.02 | 0.391 | 0.516 | 0.794 |
| 5 | 165.5 | 17.45 | 0.418 | 0.486 | 0.820 |
| 6 | 138.1 | **17.47** | **0.420** | **0.460** | **0.830** |
| 8 | **118.1** | 17.12 | 0.403 | 0.449 | 0.829 |

Table 4: Results on generation and single-view reconstruction with varying numbers of views during training, on CO3D hydrant. To allow a like-for-like comparison, we report FID and reconstruction metrics on eight held-out views, regardless of the (varying) number of views in which diffusion is performed. We see that generation uniformly improves with more training views, while reconstruction performance saturates.

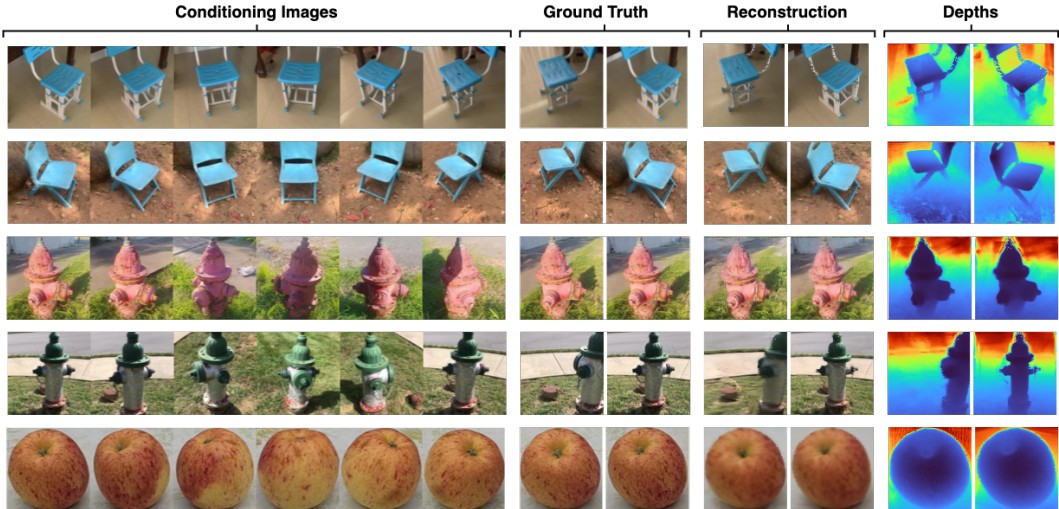

Figure 4: Results from our model on 3D reconstruction from six images. The leftmost columns are the input (conditioning) views; the next two columns show ground-truth images at novel viewpoints. The remaining columns show our model's 3D reconstruction rendered from those viewpoints, as well as the predicted depth-maps. Note how the model faithfully reconstructs the geometric and textural details visible in its input images.

### A.3 FURTHER QUALITATIVE RESULTS

In Fig. 4 we show qualitative results on reconstruction from six input views. We see that the model accurately incorporates all details of the input images. This is further shown by quantitative evaluation (Tab. 1, right four columns), showing that our model achieves high PSNR score for all datasets. In the supplementary material, we additionally include random (not cherry-picked) videos showing results from our model, to allow visualising the coherent and detailed 3D scenes it produces.

### A.4 QUALITATIVE COMPARISON WITH BASELINES

In Fig. 5 and Fig. 6, we present a qualitative comparison between our model and the preceding works PixelNeRF++ (Yu et al., 2021), Viewset Diffusion (Szymanowicz et al., 2023) and RenderDiffusion++ (Anciukevičius et al., 2023) on 3D reconstruction. Fig. 5 shows reconstruction results when the model is provided with just a single image at test time; conversely, Fig. 6 shows results when conditioned on six images. We also show unconditional generation results (without any input image) in Fig. 7. It can be seen that the prior generative approaches, RenderDiffusion++ and Viewset Diffusion, struggle to capture intricate details within scenes. On the other hand, the discriminative model PixelNeRF++ tends to render scenes that look detailed from viewpoints near the input image, but become blurry at more distant viewpoints. In contrast, our model demonstrates a superior ability to sample high-fidelity 3D scenes with greater detail and realism, especially at far-away poses, as further corroborated by the quantitative assessments in Tab. 1. We also include 2 samples (using different random seeds) of novel views produced by 2D multi-view diffusion (i.e. "no 3D" ablation of our model described in ). We

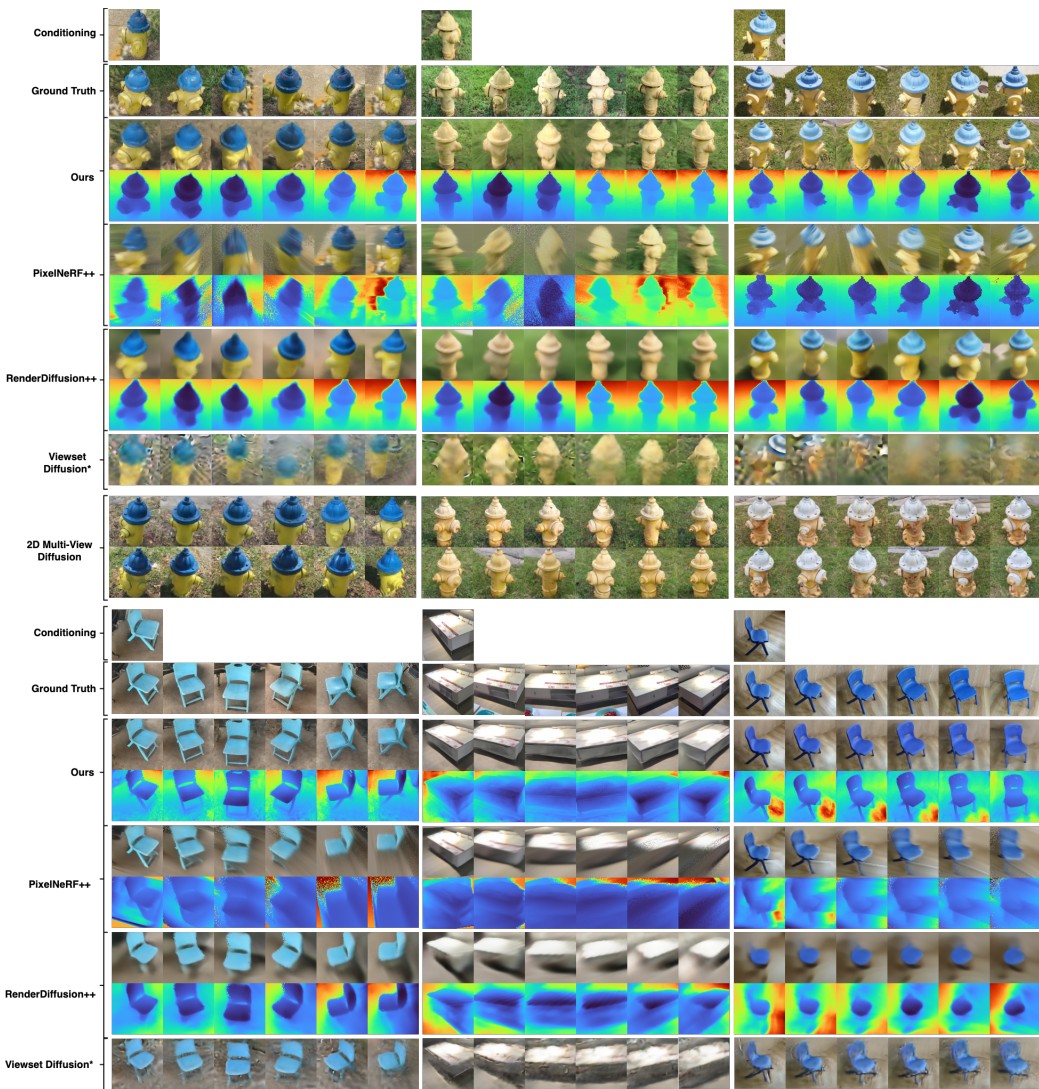

Figure 5: **Single-view 3D reconstruction** . The first row shows the conditioning (input) image, and the second row show the ground-truth for novel views. The subsequent two rows show 3D scenes sampled by our model, showing predicted views and depth maps. Corresponding results from baseline models and the 2D multi-view diffusion ablation study are also shown. Note that the 2D multi-view diffusion ablation does not generate depth maps; in this case, two rows show multi-view image samples generated using different random seeds. Our model demonstrates high-fidelity reconstruction of 3D scenes with plausible reconstructions of unseen regions. In comparison, RenderDiffusion++ samples 3D of low fidelity, while PixelNeRF++ fails render plausible details in unobserved areas. Viewset Diffusion performs well on MVImageNet, but for the larger outdoor scenes in CO3D it often renders floaters or foggy surfaces. We also see that 2D multi-view diffusion (ablation of our model) produces images that are realistic in isolation; however, they are 3D inconsistent and often do not match the ground-truth pose of the object.

see that this ablation of our model produces images that are realistic in isolation; however they are not 3D consistent and often do not match the ground-truth pose of the object.

## B ARCHITECTURE

In the following subsections we describe the details of several components, including the multi-view U-Net, feature fusion, and camera pose conditioning.

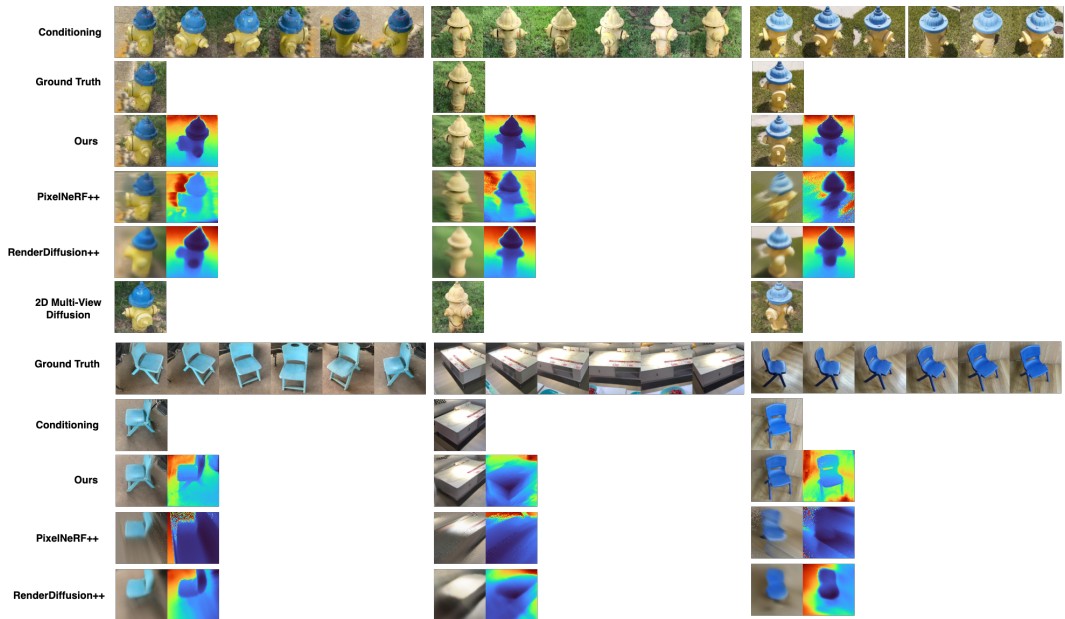

Figure 6: **Sparse-view 3D reconstruction**. The first row displays the six conditioning (input) images, and the second row presents the ground-truth for a novel view. The subsequent row depict sampled 3D scenes by our model, showing the predicted view (left) and its depth map (right). Corresponding results from baseline models and the 2D multi-view diffusion ablation study are also shown. Note that the 2D multi-view diffusion ablation does not generate depth maps; in this case, two rows show multi-view image samples generated using different random seeds. Only our model demonstrates high-fidelity reconstruction of 3D scenes.

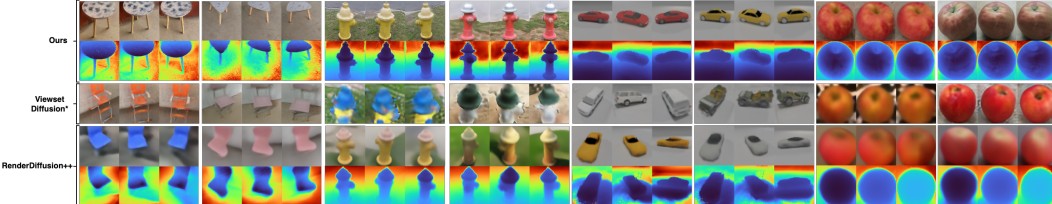

Figure 7: **Unconditional Generation Results**. Each group of 3 columns shows 3 views of a scene sampled unconditionally from the relevant model (we show 8 scenes per model). We see that our model samples the highest-fidelity scenes, especially when trained on challenging in-the-wild MVImgNet and CO3D datasets. In contrast, prior approaches (RenderDiffusion++ and Viewset Diffusion*) perform well on the simple object-centric ShapeNet dataset, but fail to synthesise fine details on MVImgNet and CO3D.

**Multi-view denoising U-Net.** We extend the denoiser's architecture to support jointly encoding varying numbers of images, to increase the quality of 3D reconstruction. Specifically, we adopt the architecture of Ho et al. (2020), which is a U-Net (Ronneberger et al., 2015) architecture with 8 ResNet blocks, each block additionally conditioned with a Fourier embedding of diffusion timestep. To support explicit 3D scene representation, we additionally condition each block on the camera pose embedding, which we get by passing the camera extrinsics and intrinsics (relative to some arbitrarily-chosen view) through a small MLP. Similarly, to support class-conditioned generation for MVImgNet and multi-class CO3D, we add a class embedding at each ResNet block. To allow the model to integrate information from arbitrarily many images, we use cross-view attention layers after each ResNet block; these are identical to typical multi-headed attention (Vaswani et al., 2017), but operate across all points in all views, by flattening these together before he attention operation.

**Multi-view feature fusion.** As described in Sec. 3.2, our multi-view encoder ingests a set of $V$ images and generates a corresponding set of $V$ feature representations. During the test phase, these features are queried at a 3D point $p \in \mathbb{R}^3$ along a some camera ray. This query is conducted via a combination of camera and equirectangular projections, mapping the 3D point $p$ into feature planes at specific camera positions. Bilinear interpolation is subsequently employed to obtain the final feature set. These features, along with the distances from the 3D points to the camera, are amalgamated using a feature fuser.

After extensive experimentation to identify the optimal architecture for this fuser, we found that multi-head attention (attending across views, independently for each 3D point $p$) out-performed other architectures, likely due to its ability to selectively focus on relevant features based on a point's distance to the camera. Despite this, our experiments also revealed that max-fusion of these features offers both faster convergence in terms of wall-clock time and robust performance with far smaller memory requirement; therefore, we report results using max-fusion in our studies. The fused feature is then passed through a 2-layer MLP to output color and density at a 3D point $p$.

**Camera pose conditioning.** To facilitate model generalization across an arbitrary number of views, we developed a conditioning approach that pools camera poses. Specifically, we employ a multi-layer perceptron (MLP) to obtain embeddings of each camera pose, each consisting of 16 extrinsic and 9 intrinsic parameters. Then to get the embedding for the i-th view, we concatenated with the max-pooling of all other viewpoint embeddings. This encapsulates not just the unique characteristics of the camera pose for the current view but also their relational context with other views.

Through various experiments, we explored multiple methods for incorporating camera pose information into our model. These approaches ranged from appending embedding as additional channels at the inception of the UNet, to including ray origin and direction as additional channels. However, our experiments indicated that introducing these camera pose embeddings at each ResNet block along with diffusion timestep conditioning yielded the best performance.

## C  TRAINING

**Optimization.** We employed the Adam (Kingma & Ba, 2015) optimizer with a learning rate of $8 \times 10^{-5}$ and beta values of $\beta_1 = 0.9$ and $\beta_2 = 0.999$ for model training. Norm-based gradient clipping was applied with a value of 1.0. We used a batch size of 8. For evaluation, we used an Exponential Moving Average (EMA) model with a decay factor of ema_decay = 0.995.

**Minibatch sampling.** To support generalization to varying numbers of images, during each training step, we randomly sample 6, 7 or 8 images from a given scene. To reduce GPU memory consumption during training, we render 12% or 5% of pixels depending on resolution.

**Volumetric rendering.** In our volumetric rendering process, each pixel was rendered by sampling 64 depths along the ray with stratified sampling, followed by 64 importance samples. We sample the background radiance from a uniform distribution when rendering each pixel.

**Denoising diffusion.** We adopt a sigmoid noise schedule (Jabri et al., 2022) with 1000 timesteps for our denoising diffusion process. To generate samples, we use 250 DDIM steps (Song et al., 2020) for unconditional generation and 50 DDIM steps for conditional generation (single-image and sparse-view reconstruction).

# D  BASELINES

We compare to the most related diffusion method RenderDiffusion (Anciukevičius et al., 2023) and the most related non-generative method PixelNeRF (Yu et al., 2021); we also compare hydrant and apple generation performance with the concurrent Viewset Diffusion (Szymanowicz et al., 2023). Like ours, RenderDiffusion performs diffusion in image space, but uses a triplane representation of latent 3D shapes and requires objects and cameras to be placed in a canonical frame of reference. PixelNeRF is not generative, but performs reconstruction from few views by unprojecting CNN features into 3D space. Viewset Diffusion is a diffusion model over masked multi-view images; however unlike ours, it represents scenes by features in a fixed-size voxel grid and so is unable to adapt its capacity nor model very large scenes.

For a fair comparison with RenderDiffusion and PixelNeRF, we adapt them to use the same feature-extraction architecture as our own work (i.e. an attentive U-Net, modified to include cross-view attention), and thus denote them as RenderDiffusion++ and PixelNeRF++. Without these modifications, both baselines fail catastrophically due to the non-canonical alignment and scales of objects and camera poses.

For Viewset Diffusion we modify their data-loader to remove masking (so the entire scene is visible, not just a single foreground object). For CO3D we retain their scene normalization so the focal object is centered in world space, upright, and of unit size – making their task considerably easier than ours. For MVImgNet, we perform similar normalization, but since the foreground object scale is unknown, we instead rescale based on the bounding box of the camera centers. For ShapeNet, we use the original world-space object and camera poses, giving Viewset Diffusion a strong advantage since all scenes have identical scale and orientation (in contrast our method only has access to the relative camera poses, and no global or object-centric frame of reference). We sweep over different values for the feature volume size, and keep other hyperparameters as in the public code. When evaluating Viewset Diffusion, we use four input views to the U-Net (as in their public code), rather than our six, and calculate heldout-view metrics on four views spaced equally between these.

# E  DATASETS

We partition each dataset into training, validation, and test sets, following a 90-5-5% split based on lexicographic ordering of provided scene names. The validation set was used for model development and hyperparameter tuning, while the test set was reserved solely for final model evaluation to mitigate any overfitting risks.

**Real-world datasets.** For the MVImgNet and CO3D datasets, the images are resized to $96 \times 96$ or $256 \times 256$ resolution. For CO3D, prior to resizing, we take a square crop centered on the ground-truth object mask, to ensure that the images are more focused on the object of interest. For MVImgNet, we take a centre crop with size equal to the smaller dimension of the image. Camera poses from COLMAP for both datasets are scaled for consistency, utilizing either the bounding box around the camera trajectory (MVImgNet) or the object point-cloud (CO3D) as a reference. Unlike Viewset Diffusion (Szymanowicz et al., 2023), we only normalize the scene scale; we do *not* modify the translation and rotation to center the object in world space, since this is not necessary with our view-space scene representation. However, when retraining Viewset Diffusion on our data, we do retain their canonicalisation of translation and rotation.

**ShapeNet.** To accelerate evaluation, we constrain the number of test scenes in the ShapeNet dataset to 150. Images in this dataset are at their original resolution of $64 \times 64$ pixels.

# F  EXPERIMENT HISTORY AND DESIGN DECISIONS

While developing the presented multi-view diffusion architecture, we have performed various failed experiments. We list these here, describing reasons for failure and our improvement, in case they are of value to other researchers.

**3D-aware single image diffusion.** Initially, we extended RenderDiffusion (Anciukevičius et al., 2023) to accommodate camera-pose-free training using an image-based rendering. We employed depth supervision to facilitate 3D reconstruction from single-image training, similar to Xiang et al. (2023). However, such design struggled to generate or reconstruct 3D assets of sufficient quality. We believe the limitations stemmed from the model not being trained to create 3D assets that are visually

compelling from multiple viewpoints. Furthermore, the 3D scene is bounded by such approach. To overcome these hurdles, we instead used multi-view image supervision and an unbounded neural scene representation, which can effectively leverage available video datasets. This enabled us to generate 3D assets that appear consistent and high-quality from various perspectives.

**Recurrent architecture.** In another line of experimentation, we explored the use of a recurrent neural network (RNN) architecture. Here, the model conditions on the prior rendering $\mathbf{u}_{i-1}$ at camera $\mathbf{c}_i$ of the scene $\mathbf{z}_{i-1}$ in a recurrent state $i$ and the noisy image $\mathbf{x}_i^{(t)}$. It is then tasked to reconstruct the image $\mathbf{x}_i$. Such a "refine, fuse and render" approach has been standard in the literature (Wiles et al., 2020; Rockwell et al., 2021). However, like our earlier single-image model, the RNN-based architecture also failed to produce high-quality 3D assets. We hypothesize that this failure is due to each recurrent state $i$ only receiving a partial observation of the scene $\mathbf{z}_{i-1}$ through its conditioning $\mathbf{c}_i$ (as it only observes camera view frustum). Hence, this again limits the model's ability to make accurate predictions for denoising the scene. Moreover, even if complete information were available, updating the scene at each 3D point would be a complex task for the model to learn. To address these issues, we transitioned to a multi-view architecture and introduced drop-out technique to enable its training. This change significantly enhanced performance, enabling the model to accurately generate and reconstruct the 3D scene with an arbitrary number of views.

**Score-distillation for scene generation.** We experimented with the score-distillation approach (Poole et al., 2022), inspired by its effectiveness in generating object-centric scenes (Wang et al., 2023). However, we found several limitations when applying it to our setting. Firstly, the method led to saturated, toy-looking scenes owing to its mode-seeking, optimization-based sampling. While this issue might not critically impact 3D generation, it becomes problematic for 3D reconstruction, which requires an accurate depiction of the conditioning image. Out-of-the-box score-distillation often fails when the conditioning image does not align with the dominant mode. Secondly, the technique necessitates careful selection of camera poses. This requirement is manageable for object-centric scenes but becomes increasingly challenging for larger, more complex scenes. Due to these limitations, we opted for classical denoising diffusion sampling of 3D scenes, similar to Anciukevičius et al. (2023). This approach is not only faster but also less memory-intensive, taking mere seconds as opposed to hours required by score-distillation.

