# OpenReview forum: "Denoising Diffusion via Image-Based Rendering"
_ICLR.cc/2024/Conference — ICLR 2024 poster_

### Official Review · Reviewer_a83P · 2023-10-22

**Soundness:** 3 good
**Presentation:** 3 good
**Contribution:** 2 fair
**Rating:** 6
**Confidence:** 5

**Summary:**

This paper proposes a new diffusion based framework that for 3D scene generation from 2D images.
The proposed method includes a new neural representation for 3D scenes, based on lifting image-based features from multiple cameras into 3D space. A denoising diffusion framework that ensures the consistency of the 3D scene throughout the generation process by first generating neural scene representations and then conduct volumetric rendering. Experiments on multiple datasets shown the proposed method could generate 3D scene with better quality than previous methods.

**Strengths:**

- The overall pipeline is clear. The idea of generating a 3D scene representation first and then render it to 2D images is reasonable for preserving 3D consistency.
- The writing is overall clear.
- The result quality looks plausible.

**Weaknesses:**

- The proposed representation of 3D scenes is not totally new. Using projected image features is common in MVS field for constructing cost volumes [W1]. Using cross-attention to fuse information from arbitrary number of views during network forward is interesting; yet in [W2], a similar network structure was also proposed to fuse features from arbitrary number of views, the only difference is they uses RNN rather than cross-attention layers. An in-depth discussion of key differences between the proposed method and these related works should be included.
- Two type of baselines is lacked for 3D generation / reconstruction: (1) Optimization-based pipeline with diffusion model as priors, e.g., [W3]; (2) Diffusion models trained on purely 3D data, e.g., [W4].
- For the visual result in supp. materials: generated results do not have comparisons. inference results do not provide input images. Lacking these materials make it hard to evaluate the quality.

[W1] Yao, Yao, et al. "Mvsnet: Depth inference for unstructured multi-view stereo." Proceedings of the European conference on computer vision (ECCV). 2018.

[W2] Riegler, Gernot, and Vladlen Koltun. "Free view synthesis." Computer Vision–ECCV 2020: 16th European Conference, Glasgow, UK, August 23–28, 2020, Proceedings, Part XIX 16. Springer International Publishing, 2020.

[W3] Liu, Ruoshi, et al. "Zero-1-to-3: Zero-shot one image to 3d object." Proceedings of the IEEE/CVF International Conference on Computer Vision. 2023.

[W4] Wang, Tengfei, et al. "Rodin: A generative model for sculpting 3d digital avatars using diffusion." Proceedings of the IEEE/CVF Conference on Computer Vision and Pattern Recognition. 2023.

**Questions:**

- In 3.1: "To ensure the scene geometry is well-defined even outside the union of the camera frustra, for each camera we also calculate a polar representation of the displacement of p relative to the camera’s center, and use the resulting angles to interpolate into a second feature map (with an equirectangular projection)" - Does the result feature map $f_v'$ is shared amount all views? How to use the angles to interpolate this feature map?
- The proposed method supports arbitrary views. How does the performance w.r.t number of views?
- Real multi-view datasets usually contain camera calibration errors (as they are usually estimated via some algorithm). How does the robustness of proposed method w.r.t camera calibrations?
- Regarding the ablation study experiment on using triplanes: It seems the experiment directly using 2D U-Net to output tri-plane features. Yet, as argued in [Q1], there exists an incompatibility between the tri-plane representation and the naive 2D U-Net method. In [Q1] a 3D-aware convolution for output tri-plane features from 2D U-Net is proposed and achieved good quality. How would this variant fit into the proposed method in this paper?

[Q1] Wang, Tengfei, et al. "Rodin: A generative model for sculpting 3d digital avatars using diffusion." Proceedings of the IEEE/CVF Conference on Computer Vision and Pattern Recognition. 2023.

---

> ### Author Response · Authors · 2023-11-15
> **Response to Reviewer a83P (1/N)**
>
> Thank you for the detailed and constructive review. Below and in the updated paper, we address your concerns and suggestions.
>
> > The proposed representation of 3D scenes is not totally new.
>
> Though image-based rendering has been studied in computer graphics, our approach makes three new technical contributions (which we now describe in section 3.1).  The first contribution (as you noted) is joint prediction of IBR features across images, via the multi-view-attentive U-Net. This is strictly more expressive than prior IBR approaches (pixelNeRF, IBRNet, FVS) that calculate features independently for each image – our multi-view U-Net can arrange different IBR features for the i-th viewpoint depending on other input images. Specifically, it can remove the ambiguity of the 3D scene that is present when given only one image (e.g. pixelNeRF, IBRNet, FVS). Prior approaches (e.g. pixelNeRF, IBRNet, FVS) had to alleviate this problem by having a large (hence expensive) model to fuse features (e.g. FVS has RNN as you mentioned whilst IBRNet has a deep attention network over point features and nearby 3D points in contrast to our simple max-pooling). Intuitively, our representation may be seen as an improvement of K-planes (https://sarafridov.github.io/K-Planes/) as (i) we arrange by placing them at camera position instead of canonical position and (ii) projecting to them using camera intrinsics instead of perpendicular projection. We have now named our representation "IB-planes" to better highlight this distinction.
>
> Second, we use a learnable embedding for sample depths, by projecting the depth of a sample point wrt each view to a learnable feature vector (in contrast to embedding global canonical position with fourier positional encoding as PixelNeRF). This was mentioned very briefly in sec. 3.1, but we will expand on this point. It allows the model to better express different features for all points along the ray, yet without requiring scenes to have a canonical orientation (as depths are relative to the camera).
>
> Third, we also define features even outside of all camera-view frustums, by including "spherical features" (i.e. polar unprojection of all points into an additional feature map for each camera). This significantly improves FID at heldout views from 126.5 to 118.1 (see “No spherical” experiment in Table 3).
>
> Together, this means our representation is more expressive than existing methods, and can represent complex in-the-wild scenes, including outdoors (see figure 4), and allows incorporating information from multi-view images for IBR-based rendering. Our experiments with triplanes (“No image-based representation” in table 3), pixelNeRF representation and ablations (e.g. w/o multi-view attention in table 3), demonstrate the superiority of our representation.
>
> However, the most important technical contribution of our work is that it is the first principled approach for integrating image-based rendering of 3D scenes into diffusion models. Mathematically, the diffusion formulation requires that all dimensions (e.g. pixels) of a noisy datapoint must be denoised by the model; changing any noise variable then correspondingly changes the sample. However, the difficulty of using diffusion with an IBR representation is that using all IBR frames, including of the (noisy) i-th image when rendering (i.e. denoising) the i-th camera pose is that it leads to pathological 3D solutions (e.g. flat planes in front of the camera). We demonstrate this with an experiment in Table 3 (“No repr. drop”) where we see that the DRC (depth rank correlation) metric worsens from 0.821 to 0.586. One naive approach to avoid trivial 3D solutions is to use held-out views for supervision. However, this approach is mathematically incorrect as the uncertainty of (or latent information about) held-out images has no connection to the noise distribution. In practice, this prevents the diffusion model from sampling plausible details in these withheld views, instead merely approximating the average observation, much like older non-generative techniques. In contrast, our model can use dropout without this limitation as the noise from each view can influence all latent 3D scene representation via multi-view attention in the UNet. We add this explanation to section 3.4.

---

> ### Author Response · Authors · 2023-11-15
> **Response to Reviewer a83P (2/N)**
>
> > Two type of baselines is lacked for 3D generation / reconstruction: (1) Optimization-based pipeline with diffusion model as priors (2) Diffusion models trained on purely 3D data
>
> We have already made significant extensions to state-of-the-art works (e.g. RenderDiffusion) to make them support our challenging in-the-wild setting. Nevertheless, we agree it is valuable to include an “Optimization-based pipeline with diffusion model as priors”. We therefore now include results from SparseFusion on the CO3D Hydrant dataset (we will add results on other datasets soon). We see that SparseFusion performs significantly worse than our approach, and is also far slower than ours. Specifically, SparseFusion achieves PSNR of 12.06 (vs **16.07** for ours) and LPIPS of 0.63 (vs **0.456** ours). As we explain in sec. 2, this is because score-distillation is not a good objective for reconstruction due to its mode-seeking behaviour – this fails when the conditioning image does not align with the dominant mode. We also already include an experiment with generating multi-view images using only 2D multi-view diffusion. However, in contrast to Zero-1-to-3, which samples multi-view images independently, we train this 2D baseline to sample multi-view images jointly, using cross-view attention so the model is expressive enough to learn 3D consistency. Our experiments in Table 3 (row “No 3D”) demonstrate that resulting reconstructed images score poorly according to PSNR & SSIM, since they fail to show a coherent and accurate 3D scene (LPIPS is slightly better since it is less sensitive to fine details in the images).
>
> Regarding Rodin as an additional baseline, we have already included an experiment with triplanes instead of our IB-planes representation (see “No image-based representation” paragraph in A.1). Note that Rodin itself would require very substantial modifications and extensions to work in our setting. In particular, Rodin is designed to work with 1) masked objects 2) a bounded scene volume 3) canonically aligned camera poses, e.g. all faces in the dataset point north in a shared coordinate frame 4) uniform scales (i.e. facial features of all people in the dataset are of nearly identical size). These assumptions are reflected in their use of canonically-aligned triplanes (in world-space) as a 3D representation; we already note (in App. D) that this representation performs very poorly in our more challenging setting. In contrast, our in-the-wild datasets (e.g. MVImgnet) contain arbitrary scene scales and orientations, diverse camera poses, and large-scale outdoor scenes. Nevertheless, if you still feel that we should include a baseline of this kind (regardless of the very different setting and of our existing “No image-based representation” experiment results), we are happy to implement and test a suitable variant of our model (since there is no public implementation of Rodin) if you could let us know what you would consider a fair comparison:
> * Which representation to use – e.g. triplanes in canonical world-space orientation or in view-space?
> * How to condition the diffusion model on multi-view images and their camera poses?
> * How to rescale the scenes (which have arbitrary scales from COLMAP) so that they are modelled by a fixed-size triplane/voxel-grid?
>
>
> > Missing baseline qualitative results in the supplementary
>
> Thank you for noting this omission! Today, we will include visualisations in the supplementary material for pixelNeRF, RenderDiffusion and Viewset-Diffusion. We will also release corresponding video visualisations on the project website upon acceptance.

---

> > ### Comment · Reviewer_a83P · 2023-11-18
> >
> > > Nevertheless, we agree it is valuable to include an “Optimization-based pipeline with diffusion model as priors”. We therefore now include results from SparseFusion on the CO3D Hydrant dataset (we will add results on other datasets soon).
> >
> > Thanks for adding the comparisons. It makes the experiment more convincing.
> >
> > > Nevertheless, if you still feel that we should include a baseline of this kind (regardless of the very different setting and of our existing “No image-based representation” experiment results), we are happy to implement and test a suitable variant of our model (since there is no public implementation of Rodin) if you could let us know what you would consider a fair comparison.
> >
> > The original purpose of my proposal for compare with Rodin is to compare with some methods that purely trained with 3D data, i.e., they have the maximized accurate 3D information from data itself. Yet I agree with you that Rodin was dedicated designed for specific subspaces (i.e., human and/or faces). Hence, I would say that (1) it's ok to not compare with Rodin, and (2) I think the discussion of Rodin (as in your response) is reasonable and maybe it can be briefly included in the final version.
> >
> > > Thank you for noting this omission! Today, we will include visualisations in the supplementary material for pixelNeRF, RenderDiffusion and Viewset-Diffusion. We will also release corresponding video visualisations on the project website upon acceptance.
> >
> > That's great :)

---

> ### Author Response · Authors · 2023-11-15
> **Response to Reviewer a83P (N/N)**
>
> > “inference results do not provide input images.”
>
> We do already show the input images in fig. 3, in the leftmost column titled "conditioning". However, we will add an explanation in the caption.
>
> > Is the (polar) feature map shared across all views? How to use the angles to interpolate this feature map?
>
> No, one feature map is predicted for each view, as a second output from the U-Net, in the same way as we predict feature planes in each view frustum. This can be thought as warping a 2D feature plane into a sphere using equirectangular projection, i.e. we use the (normalised) polar coordinates theta and phi to choose where to take these features from.
>
> > How does performance vary w.r.t number of views?
>
> We already include detailed experiments to answer this question. On one hand, increasing the number of noisy views during training leads to better performance across both generation and reconstruction metrics (see Table 4 and sec. A.2). However, increasing the number of views has diminishing returns. Specifically, increasing from 3 to 5 improves FID by 58.4, while increasing from 6 to 8 improves FID by 20.0. On the other hand, increasing the number of conditioning views during test-time reconstruction naturally increases the fidelity of the scene as there is less uncertainty. Specifically, increasing the number of input views from 1 to 6 increases the PSNR from 18.54 to ​​22.09 on MVImgNet (see Table 1).
>
> > Real multi-view datasets usually contain camera calibration errors (as they are usually estimated via some algorithm). How does the robustness of proposed method w.r.t camera calibrations?
>
> Indeed, 4 out of our 5 datasets contain significant camera calibration errors. For example, the creators of CO3D state that around 18% of camera poses are inaccurate. We do not filter these as we can see that our model is robust to these errors. However, we do notice increased quality for more accurate camera poses (e.g. classes with accurate camera poses perform better than without; however, we have not quantified this error as it is difficult to come up with a measure).
>
> > It seems the experiment directly using 2D U-Net to output tri-plane features. Yet, as argued in [Q1], there exists an incompatibility between the tri-plane representation and the naive 2D U-Net method. In [Q1] a 3D-aware convolution for output tri-plane features from 2D U-Net is proposed and achieved good quality. How would this variant fit into the proposed method in this paper?
>
> Thanks for pointing this out. We too conducted some initial experiments with a more sophisticated unprojection operation. However, we observed that our IB-planes representation performed well without the additional compute and memory requirements for a 3D-aware convolution. We concluded that such techniques are only needed when the mapping from image pixels into triplane features is difficult (as also shown by recent works), which is not the case for our IB-planes representation as it instead defines and exploits a simple mapping from the image to IBR features, which is both fast and expressive.

---

> > ### Comment · Reviewer_a83P · 2023-11-18
> >
> > Thanks for the detailed response.
> >
> > TL; DR - this response clarifies many things, and I would raise my score from 5->6.
> > Here are my further comments (with still some questions) as follows:
> >
> > > This is strictly more expressive than prior IBR approaches (pixelNeRF, IBRNet, FVS) that calculate features independently for each image
> >
> > Agreed.
> >
> > > Prior approaches (e.g. pixelNeRF, IBRNet, FVS) had to alleviate this problem by having a large (hence expensive) model to fuse features (e.g. FVS has RNN as you mentioned whilst IBRNet has a deep attention network over point features and nearby 3D points in contrast to our simple max-pooling)
> >
> > Regarding FVS - FVS uses a UNet with GRU to fuse features before fusing them with weighted sum; the proposed method uses a UNet with attention to fuse features and fusing them with max-pooling. I still somehow think intrinsically they are similar with "fusing multiple views". Yet I agree the proposed method are quite different in implementations, and in a completely different setting with diffusion models.
> >
> > > Intuitively, our representation may be seen as an improvement of K-planes (https://sarafridov.github.io/K-Planes/) as (i) we arrange by placing them at camera position instead of canonical position and (ii) projecting to them using camera intrinsics instead of perpendicular projection. We have now named our representation "IB-planes" to better highlight this distinction.
> >
> > The relationship to K-planes (actually tri-planes since the proposed method does not handle dynamic inputs yet) seems interesting. K-planes also handles dynamic scene by having additional t-planes (which are projected "perpendicularly" in some sense similar to other spatial axis). A very interesting future avenue would be extending IB-planes for 4D videos.
> >
> > > Second, we use a learnable embedding for sample depths, by projecting the depth of a sample point wrt each view to a learnable feature vector (in contrast to embedding global canonical position with fourier positional encoding as PixelNeRF). This was mentioned very briefly in sec. 3.1, but we will expand on this point. It allows the model to better express different features for all points along the ray, yet without requiring scenes to have a canonical orientation (as depths are relative to the camera).
> >
> > My understanding is that this is like a "learnable position encoding" for depth values. Is my understanding correct?
> >
> > > Third, we also define features even outside of all camera-view frustums, by including "spherical features" (i.e. polar unprojection of all points into an additional feature map for each camera). This significantly improves FID at heldout views from 126.5 to 118.1 (see “No spherical” experiment in Table 3).
> >
> > Does the spherical features including unprojection of all 3D points, or only the projected 2D points (in pixel space)?
> >
> > > However, the most important technical contribution of our work is that it is the first principled approach for integrating image-based rendering of 3D scenes into diffusion models.
> >
> > Very interesting point. If my understanding is correct, during training, the proposed method (1) conducts a full diffusion with cross-attention with all views to produce a feature for each training view, and then (2) render each view with the feature of current rendered view discarded? Also, do you always discard the self-view, or it would be discarded with some probability?

---

> > > ### Author Response · Authors · 2023-11-19
> > > **Response to additional questions raised by Reviewer a83P**
> > >
> > > > A very interesting future avenue would be extending IB-planes for 4D videos.
> > >
> > > Thanks for the interesting suggestion! We will mention this as possible future work
> > >
> > > > My understanding is that this is like a "learnable position encoding" for depth values. Is my understanding correct?
> > >
> > > Yes, this is correct.
> > >
> > > > Does the spherical features including unprojection of all 3D points, or only the projected 2D points (in pixel space)?
> > >
> > > For every 3D point rendered (i.e. visible in some output image), we map it to 2D feature space twice per source (IBR) viewpoint – once by regular perspective projection (to sample the main image-aligned features), and once by equirectangular projection (to sample the additional spherical features).
> > >
> > > > Does it mean that the [spherical] features are re-sampled by assuming it is on an infinite-far plane?
> > >
> > > These additional features are sampled as if placed on an infinitely-distant sphere (i.e. a sky-sphere), with equirectangular projection into the 2D feature map. So they are defined even outside the view frustum for a given viewpoint, although they are still predicted based on the noisy image at that viewpoint.
> > >
> > > > Very interesting point. If my understanding is correct, during training, the proposed method (1) conducts a full diffusion with cross-attention with all views to produce a feature for each training view, and then (2) render each view with the feature of current rendered view discarded? Also, do you always discard the self-view, or it would be discarded with some probability?
> > >
> > > Yes, this is correct. We always discard the self-view. This is why we claim that this work presents the first principled approach for integrating image-based rendering into diffusion models as: (1) we discard the self-view to avoid trivial 3D; but (2) our U-Net uses cross-view attention that integrates the noise (and thus latent information) from all views to predict each IB-plane.
> > >
> > > >  Yet I agree with you that Rodin was dedicated designed for specific subspaces (i.e., human and/or faces). Hence, I would say that (1) it's ok to not compare with Rodin, and (2) I think the discussion of Rodin (as in your response) is reasonable and maybe it can be briefly included in the final version.
> > >
> > > Thank you, we now added an additional sentence to the second paragraph in the Related Work section.
> > >
> > > ***Thank you again for the particularly constructive feedback! We hope we have now addressed all your questions. Let us know if you have any further concerns.***

---

> ### Comment · Reviewer_a83P · 2023-11-18
>
> > We do already show the input images in fig. 3, in the leftmost column titled "conditioning". However, we will add an explanation in the caption.
>
> Thanks for the clarification.
>
> > No, one feature map is predicted for each view, as a second output from the U-Net, in the same way as we predict feature planes in each view frustum. This can be thought as warping a 2D feature plane into a sphere using equirectangular projection, i.e. we use the (normalised) polar coordinates theta and phi to choose where to take these features from.
>
> Does it mean that the features are re-sampled by assuming it is on an infinite-far plane?
>
> > We already include detailed experiments to answer this question (performance vary w.r.t number of views).
>
> Now I see the Table 4 and Sec A.2. Thanks for the clarification.
>
> > For example, the creators of CO3D state that around 18% of camera poses are inaccurate. We do not filter these as we can see that our model is robust to these errors.
>
> Thanks for the clarification.

---

### Official Review · Reviewer_Wz4z · 2023-10-29

**Soundness:** 3 good
**Presentation:** 3 good
**Contribution:** 3 good
**Rating:** 8
**Confidence:** 3

**Summary:**

This paper proposes to use a multi-view version of a diffusion model for 3D reconstruction, which would be suitable for estimating occluded parts from a relatively small number of viewpoints. The proposed method involves several technical novelty, in which the key technical contribution may be in the methods of 1) NeRF-like 3D representation using image features and 2) exploiting multi-view consistency to diffusion models. The experiments show that the proposed method achieves better novel-view synthesis accuracies compared with, e.g., RenderDiffusion and PixelNeRF.

**Strengths:**

+ For the context of usual multi-view 3D reconstruction, in which the camera views are not structured (i.e., allowing free movement), this paper may be the first attempt to use modern image generation methods like diffusion models.
+ The performance of the proposed method outperforms SOTA diffusion-based 3D reconstruction.
+ The simple regularization used in this work, dropping out the features from a view when rendering to that same viewpoint, would be helpful in a broader context.

**Weaknesses:**

- There are some unclear technical details and contributions (see detailed comments).

- As a single-view 3D reconstruction method, I agree with the practical value of using diffusion models or similar stuff. However, the proposed method basically targets the multi-view contexts. For multi-view 3D reconstruction, usual MVS-based methods (or neural-MVS-based methods, perhaps) may still achieve much better performance for large-scale and detailed reconstruction.
A fundamental drawback of using multi-view-consistent generative models for multi-view 3D reconstruction may be in the limitation of input image resolution (due to the size of graphics memory), as well as the generative models change the image content that can degrade the faithfulness of the 3D reconstruction.

- Related to the above, a potential merit of using generative models for 3D reconstruction is the occlusion recovery, as written in the introduction. In such senses, I could see experiments assessing the results of 3D reconstruction under severe occlusions.

- Also related to the above, a potential drawback of using diffusion models for 3D reconstruction may be that they change the content of given multi-view images. I could see some discussions (and potential failure cases, if any).

**Questions:**

- The contribution statement is not clear: "first denoising diffusion model that can generate and reconstruct large-scale and detailed 3D scenes." Which part is actually the "first"? Which adjective corresponds to which technical part? For example, readers may think RenderDiffusion met the above stuff.

- Related to the above, it is pretty helpful to state the technical difference and practical merit compared with recent methods like RenderDiffusion and PixelNeRF.

- As the paper mentions, the proposed 3D representation is similar to PixelNeRF. To the 3D representation, which part is the key technical difference from PixelNeRF?

- The multi-view version of diffusion models inputs the camera poses as embeddings. If allowing the arbitrary viewpoints to reconstruction, I thought it is not enough just to input (and map) camera matrices to achieve reasonable multi-view conditioning on diffusion models. The multi-view image features are just concatenated (i.e., pixel-aligned), so there should be a part that computes correspondence among the views. Attention layers may do it, but I am wondering if it is really capable of finding correct correspondences from the given information (without, e.g., geometrically projecting the pixels to 3D scenes during the training of diffusion models). Do they use any strong assumptions, e.g., camera locations are almost the same or less variety for training datasets?

---

> ### Author Response · Authors · 2023-11-15
> **Response to Reviewer Wz4z (1/N)**
>
> We are grateful for your constructive feedback, thanks to which we improved the clarity of the paper by incorporating your suggestions and performing additional experiments. Below we address one misunderstanding (and underestimation) of our work and answer your other questions.
>
> > the proposed method basically targets the multi-view contexts. For multi-view 3D reconstruction, usual MVS-based methods (or neural-MVS-based methods, perhaps) may still achieve much better performance for large-scale and detailed reconstruction.
>
> This is incorrect, we specifically target the more challenging tasks of (i) unconditional 3D generation (ii) 3D reconstruction from a single view and (iii) 3D reconstruction from very few (6) views. Even during training we only require at most 8 views per scene. In contrast, MVS-based methods (and NeRFs) typically require more than 20 images. Another crucial disadvantage of MVS-based methods is that they are not generative, so they cannot generate plausible details in parts of the scene that are unobserved (e.g. due to occlusion). In contrast, our approach can sample multiple plausible completions of unobserved regions, that are compatible with the input image. We now added further clarifications in sec. 1 (1st paragraph) and sec. 2 (1st paragraph).
>
> > A fundamental drawback of using multi-view-consistent generative models for multi-view 3D reconstruction may be in the limitation of input image resolution (due to the size of graphics memory), as well as the generative models change the image content that can degrade the faithfulness of the 3D reconstruction.
>
> Regarding faithfulness of the reconstruction, a particular benefit of our IBR-based scene representation is that it can preserve fine details in image-space, avoiding a mapping to a smaller latent space that might indeed remove such details. This is evidence in fig. 3, where very fine details (e.g. narrow chair legs) are preserved in the reconstructions.
> Regarding resolution, our model estimates a continuous 3D scene, and we can render its samples at any resolution. Indeed, we have now experimented with rendering reconstructions at 1024x1024 resolution. We see that it performs well, predicting additional details not visible in the input 96x96 image. Hence, our model can also perform superresolution by first predicting a 3D asset from a conditioning image and then rendering the 3D at a higher resolution. It is noteworthy that this is the highest resolution supported by any generative 3D work to date; we now state this contribution explicitly in the introduction.
>
> > a potential merit of using generative models for 3D reconstruction is the occlusion recovery, as written in the introduction. In such senses, I could see experiments assessing the results of 3D reconstruction under severe occlusions.
>
> Again, we already report extensive experiments on single-image 3D reconstruction, where the model is tasked to generate occluded regions, notably the reverse of the object and volumes outside of the input image (both visible in 360deg spin views around the object). We always see that our model samples plausible 3D completions of occluded volumes, for example, in Figure 3 row 4, you can see that our model can take as input the single image on the left (column name ‘conditioning’) and output a full 3D scene, whose renderings we visualise as “Reconstruction”. We see that our model correctly predicts the shape of the hydrant (including the back), as it has learned a prior over 3D scenes (e.g. that hydrants tend to be cylindrical with pipes at the top). Also, note that the model also samples a plausible background (flat, with green grass around it) for the region behind the original camera, despite it not being observed in the original conditioning image. We now emphasise this further in the experiments section 4.1.

---

> ### Author Response · Authors · 2023-11-15
> **Response to Reviewer Wz4z (2/N)**
>
> > Also related to the above, a potential drawback of using diffusion models for 3D reconstruction may be that they change the content of given multi-view images. I could see some discussions (and potential failure cases, if any).
>
> One of the major strengths of our work is that the model faithfully represents the input/conditioning images -- this is ensured by our neural 3D scene representation (section 3.1) which projects rendered 3D points to all images to query image features, which are then fused to output radiances. This is in contrast to prior works, such as RenderDiffusion, which requires black-box CNN to map to 3D features. We now emphasised this point further in section 3.1.
>
> > The contribution statement is not clear: "first denoising diffusion model that can generate and reconstruct large-scale and detailed 3D scenes." Which part is actually the "first"? Which adjective corresponds to which technical part? For example, readers may think RenderDiffusion met the above stuff. Related to the above, it is pretty helpful to state the technical difference and practical merit compared with recent methods like RenderDiffusion and PixelNeRF.
>
> Thanks, we have now clarified these points (including our contributions) in sec. 1 & 2. Specifically, we refer to the first method capable of reconstructing large in-the-wild 3D scenes from a single image. As we explain in the Related Work section, prior work, including pixelNeRF, cannot achieve plausible reconstruction in unobserved/ambiguous regions as this requires a generative capability that they lack. On the other hand, generative models, e.g. RenderDiffusion, also had some major limiting assumptions that prevented them from achieving this task. For example RenderDiffusion has a limited 3D representation that can only represent objects but not scenes and requires canonically oriented camera poses only available in synthetic datasets. This statement is verified by reconstruction experiments in 4.1, which show very significant improvement over all prior works. We now include specific contributions in the introduction as well as specify additional technical contributions in sections 3.1 and 3.2.
>
> > As the paper mentions, the proposed 3D representation is similar to PixelNeRF. To the 3D representation, which part is the key technical difference from PixelNeRF?
>
> Thanks to this question, we have now rewrote section 3.1 to include specific technical contributions. To summarise, our approach makes three new technical contributions.
>
> The first contribution is joint prediction of IBR features across images, via the multi-view-attentive U-Net. This is strictly more expressive than prior IBR approaches (pixelNeRF, IBRNet) that calculate features independently for each image – our multi-view U-Net can arrange different IBR features for the i-th viewpoint depending on other input images. Specifically, it can remove the ambiguity of the 3D scene that is present when given only one image (e.g. pixelNeRF, IBRNet). Prior approaches (e.g. pixelNeRF, IBRNet) had to alleviate this problem by having a large (hence expensive) model to fuse features (e.g. IBRNet has a deep attention network over point features and nearby 3D points in contrast to our simple max-pooling). Intuitively, our representation may be seen as an improvement of K-planes (https://sarafridov.github.io/K-Planes/) as (i) we arrange by placing them at camera position instead of canonical position and (ii) projecting to them using camera intrinsics instead of perpendicular projection. We have now named our representation "IB-planes" to better highlight this distinction.
>
> Second, we use a learnable embedding for sample depths, by projecting the depth of a sample point wrt each view to a learnable feature vector (in contrast to embedding global canonical position with fourier positional encoding as PixelNeRF). This was mentioned very briefly in sec. 3.1, but we will expand on this point. It allows the model to better express different features for all points along the ray, without requiring scenes to have a canonical orientation (as depths are relative to the camera).
>
> Third, we also define features even outside of all camera-view frustums, by including "spherical features" (i.e. polar unprojection of all points into an additional feature map for each camera). This significantly improves FID at heldout views from 126.5 to 118.1 (see “No spherical” experiment in Table 3).
>
> Together, this means our representation is more expressive than existing methods, and can represent complex in-the-wild scenes, including outdoors (see figure 4), and allows incorporating information from multi-view images for IBR-based rendering. Our experiments with triplanes (“No image-based representation” in table 3), pixelNeRF representation and ablations (e.g. w/o multi-view attention in table 3), demonstrate the superiority of our representation.

---

> ### Author Response · Authors · 2023-11-15
> **Response to Reviewer Wz4z (N/N)**
>
> > The multi-view version of diffusion models inputs the camera poses as embeddings. If allowing the arbitrary viewpoints to reconstruction, I thought it is not enough just to input (and map) camera matrices to achieve reasonable multi-view conditioning on diffusion models. […] Do they use any strong assumptions, e.g., camera locations are almost the same or less variety for training datasets?
>
> Indeed, our experiments (called “No 3D” in Table 3) show that it is not sufficient to condition the diffusion model on camera poses without geometric priors. Hence, one of our technical contributions is to resolve this by leveraging the geometric correspondence between image-space features and 3D points. We have now emphasised this more in section 3.1 and 3.2. We do not make any strong assumptions on the diversity of camera poses in the training set – our model is trained on challenging real-world collections of images (e.g. MVImgNet), without using any masks or depths; this we now emphasise at the start of experiments section.

---

> ### Comment · Reviewer_Wz4z · 2023-11-22
> **Thanks for the authors' comments!**
>
> Thanks for the authors' detailed responses.
> I think most concerns of mine were addressed; these would be beneficial for clarifying the contribution of this work.
> Especially, the description of the proposed method as an extension from K-planes might give a good intuition for readers.

---

### Official Review · Reviewer_C6XK · 2023-10-30

**Soundness:** 3 good
**Presentation:** 2 fair
**Contribution:** 3 good
**Rating:** 6
**Confidence:** 3

**Summary:**

The paper introduces a diffusion model for generating new perspectives of reconstructed 3D scene using 2D inputs from various viewpoints. The authors employ a set of techniques, such as a 3D feature representation and feature dropping during training, to enhance the consistency of the 2D diffusion model in view synthesis. They also extend the model's capability to accommodate different numbers of input views, making it more versatile. The experimental results presented by the authors demonstrate superior performance compared to previous approaches.

**Strengths:**

This paper presents a diffusion-based model for handling the novel-view generation problem given several inputs of viewpoints. This model is able to generate new content in areas where the given viewpoints are not covered.

**Weaknesses:**

I have two main concerns about this paper:

- 1. After reviewing the supplementary material, I noticed that the generated images from different viewpoints don't seem very consistent. The videos display noticeable bouncing. Have the authors conducted both qualitative and quantitative assessments to verify if the proposed representation approach truly maintains consistency among the outputs of the diffusion model for the same 3D scene?

- 2. The paper presentation could be enhanced. While Figure 1 demonstrates the adaptive neural scene representation, there is no figure of the structure or pipeline of the multi-view diffusion model. Given that this diffusion model plays a crucial role in this paper and contains modules like the setwise multi-view encoder, it deserve a figure. Otherwise, readers might find it difficult to quickly grasp the authors' intention.

**Questions:**

As mentioned in the weaknesses section, have the authors undertaken both qualitative and quantitative evaluations to figure out if their method indeed preserves consistency? (or at least an ablation study)

While I acknowledge that perfection might not be attainable due to the presence of the black-box diffusion model, an examination in this regard is necessary to substantiate the authors' assertion that *their method ensure 3D consistency while retaining expressiveness*.

---

> ### Author Response · Authors · 2023-11-15
> **Response to Reviewer C6XK**
>
> Thank you for the positive reception of our work! We have now fixed the weaknesses you mentioned.
>
> > After reviewing the supplementary material, I noticed that the generated images from different viewpoints don't seem very consistent. The videos display noticeable bouncing.
>
> This is purely a visualisation problem, which arose due to inconsistent cropping. We have now fixed this to stay consistent throughout the video. We invite you to see the updated supplementary videos, which clearly show the multi-view 3D consistency of the generated images.
>
> > The paper presentation could be enhanced. While Figure 1 demonstrates the adaptive neural scene representation, there is no figure of the structure or pipeline of the multi-view diffusion model. Given that this diffusion model plays a crucial role in this paper and contains modules like the setwise multi-view encoder, it deserve a figure.
>
> Thank you, we now added an additional figure to ​​supplementary section B to also show our diffusion architecture and its connection with the scene representation.
>
> > have the authors undertaken both qualitative and quantitative evaluations to figure out if their method indeed preserves consistency? (or at least an ablation study). While I acknowledge that perfection might not be attainable due to the presence of the black-box diffusion model, an examination in this regard is necessary to substantiate the authors' assertion that their method ensure 3D consistency while retaining expressiveness.
>
> We emphasise that our neural scene representation defines an explicit 3D scene (i.e. a density and color at every point in 3D space), that is rendered directly to the output images as the last stage of the denoiser model. Thus, 3D scenes generated by our method are guaranteed to be 3D-consistent when rendered from any viewpoint. This is in contrast to neural rendering methods that use implicit features and black-box postprocessing – it is a particular benefit of our model that the denoiser is not a black box. We have now further clarified this in section 3.1 and 3.2.

---

> > ### Comment · Reviewer_C6XK · 2023-11-22
> >
> > Thanks for the clarification! The response by the authors has addressed my previous concern.
> > The recently updated videos, such as the #7 and #8 videos, exhibit better stability when compared to the earlier results. However, in other samples, there is still some bouncing, which I'm not sure is solely caused by a visualization issue. I tend to maintain the current rating.

---

> > > ### Author Response · Authors · 2023-11-22
> > >
> > > Thank you for the constructive feedback! We are happy to hear we addressed your concerns. Regarding some bouncing remaining in the videos, we again emphasise that explicit 3D representations allow rendering at any pose and resolution, for visualisation purposes we render to randomly-selected real-world trajectories from the CO3D dataset, where the person is walking with a camera around the object. Depending on the camera stabilisation system, the camera trajectory may contain some bouncing. We can fix this in the camera-ready version by stabilising extrinsics, if you feel this is necessary. Again, this is purely a visualisation problem.

---

> > > > ### Comment · Reviewer_C6XK · 2023-11-22
> > > >
> > > > Thank you again for the detailed explanation. Given that reason, I think the bouncing in the video is acceptable.

---

### Official Review · Reviewer_MQQ6 · 2023-11-05

**Soundness:** 2 fair
**Presentation:** 3 good
**Contribution:** 2 fair
**Rating:** 6
**Confidence:** 4

**Summary:**

This paper presents a diffusion model for fast and detailed reconstruction and generation of realistic 3D scenes. It introduces a new neural scene representation, a denoising-diffusion framework for learning a prior over the representation, and a unified architecture for conditioning on varying numbers of images. This method achieves SOTA performance in 3D reconstruction and unconditional generation according to the experiments part.

**Strengths:**

1. This method achieves state-of-the-art performance in both 3D reconstruction and unconditional generation tasks. Notably, in Figure 2, impressive scene-level 3D generation results are observed, surpassing the capabilities of prior works.
2. The designed method is practical as it can accommodate arbitrary numbers of input views, making it versatile and applicable in various scenarios.
3. The methodology section is characterized by its clarity and accessibility, facilitating a comprehensive understanding of the entire design.

**Weaknesses:**

1. No qualitative comparison was made with baselines, particularly VSD. While this paper includes numerous quantitative comparisons with three baselines, the figures only display the results of this paper in recontruction and generation task. The visual quality of VSD is also good, and it would be valuable to include its results for comparison.
2. No quantitative and qualitative comparison was made with VSD in the task of single view reconstruction.
3. The experiments section lacks clarity, as the pixelnerf++ and renderdiff++ are not explained in the main paper but rather in the appendix. Additionally, although the authors revised VSD, the table still refers to it as VSD instead of VSD*, and this should be explicitly noted.

**Questions:**

The experiments section requires improvement, specifically in the reconstruction task where it is important to include a comparison with VSD. Additionally, qualitative comparisons with all three baselines in both tasks should be provided. It is crucial to include the ablation study in the main paper rather than relegating it to the appendix. Overall, the experiments section needs reorganization to better incorporate these essential components.

---

> ### Author Response · Authors · 2023-11-15
> **Response to Reviewer MQQ6**
>
> Thank you for the positive reception of our work and for the constructive and specific suggestions – we hope we have taken all of them into account in the revised version. Below we address some remaining concerns you raised. We also invite you to look at the additional points we now emphasise thanks to the other reviewers, including specific novelties of our neural representation and the ability to render 3D scenes at 1024x1024 resolution.
>
> > No qualitative comparison was made with baselines, particularly VSD.
>
> Thanks for noticing this omission! We will today include qualitative comparison with VSD, pixelNeRF++ and RenderDiffusion++ in the supplementary.
>
> > No quantitative and qualitative comparison was made with VSD in the task of single view reconstruction.
>
> Again, thanks for noting this omission. We are running this evaluation now, and will add the results in a couple of days.
>
> > The experiments section lacks clarity, as the pixelnerf++ and renderdiff++ are not explained in the main paper but rather in the appendix. Additionally, although the authors revised VSD, the table still refers to it as VSD instead of VSD*, and this should be explicitly noted.
>
> Thanks, we now denote VSD as VSD* -- though note that we only relaxed their assumption of masks and increased the voxel grid size. We also have now moved the explanation of baselines to the main paper, in addition to the explanation in the supplementary.
>
> > It is crucial to include the ablation study in the main paper rather than relegating it to the appendix.
>
> Thank you, we have now moved the ablation study to the main paper.

---

### Author Response · Authors · 2023-11-15
**Common response and updates for revision**

We thank all four reviewers for the positive and detailed feedback. We are particularly grateful for the comments that led us to run some additional experiments demonstrating benefits of our model (support for rendering at very high resolutions), and comments that helped us clarify the novelties in our new neural scene representation, emphasising important contributions versus prior works. Finally, we hope we addressed all concerns raised by reviewers, including the following improvements:
* [a83P, MQQ6] We will now include visual comparisons to baselines in the supplementary material
* [a83P] We now include a comparison to SparseFusion on the CO3D dataset (showing our method performs significantly better), with remaining datasets being updated now or in the camera ready version
* [C6XK] We have fixed the video visualisations to better show 3D-consistency, which is in fact guaranteed by our explicit volumetric representation. Please see supplementary videos.
* [C6XK, MQQ6, Wz4z] We have improved the presentation of some sections. We now describe the experimental setup more precisely, discuss our model’s ability to generate occluded regions, specified the technical contributions over prior IBR representations and over prior generative models, moved the ablation experiments to the main paper, and explained how our model guarantees 3D consistency.

We provided detailed responses to each reviewer below.

---

### Meta-Review · Area_Chair_8KxG · 2023-12-09

**Metareview:**

This paper proposed 3D scene reconstruction method by proposing a new scene representation method, incorporating diffusion model as a prior for the 3D scene representation with strengths in generating consistent texture in occluded region.

Reviewers recognized the strengths in the capability of generating 3d scenes from unstructured camera views and the simplicity of regularization. The novelty of the 3D representation and the lack of comparisons were main concerns, however, they were resolved during the author-reviewer discussions.

Considering the strenghts and the resolved weaknesses, this paper is recommended to be accepted.

**Justification For Why Not Higher Score:**

Although the major concerns from the reviewers initial comments were resolved, the overall rating were not very high. Also, the authors had to request several qualitative & quantitative comparisons to make the experimental results valid.

**Justification For Why Not Lower Score:**

While the it took reviewers's efforts to improve the paper, the authors addressed the concerns which makes the strengths of the paper outweigh the initial drawbacks.

---

### Decision · Program_Chairs · 2024-01-16

Accept (poster)